# H2OFlow: Grounding Human-Object Affordances with 3D Generative Models and Dense Diffused Flows

**Harry Zhang**
MIT
Cambridge, MA 02139
harryz@mit.edu

**Luca Carlone**
MIT
Cambridge, MA 02139
lcarlone@mit.edu

## Abstract

Understanding how humans interact with the surrounding environment, and specifically reasoning about object interactions and affordances, is a critical challenge in computer vision, robotics, and AI. Current approaches often depend on labor-intensive, hand-labeled datasets capturing real-world or simulated human-object interaction (HOI) tasks, which are costly and time-consuming to produce. Furthermore, most existing methods for 3D affordance understanding are limited to contact-based analysis, neglecting other essential aspects of human-object interactions, such as orientation (*e.g.*, humans might have a preferential orientation with respect certain objects, such as a TV) and spatial occupancy (*e.g.*, humans are more likely to occupy certain regions around an object, like the front of a microwave rather than its back). To address these limitations, we introduce *H2OFlow*, a novel framework that comprehensively learns 3D HOI affordances — encompassing contact, orientation, and spatial occupancy— using only synthetic data generated from 3D generative models. H2OFlow employs a dense 3D-flow-based representation, learned through a dense diffusion process operating on point clouds. This learned flow enables the discovery of rich 3D affordances without the need for human annotations. Through extensive quantitative and qualitative evaluations, we demonstrate that H2OFlow generalizes effectively to real-world objects and surpasses prior methods that rely on manual annotations or mesh-based representations in modeling 3D affordance.

## 1 Introduction

The rapid advancement of AI and robotics demands next-generation agents that can perceive and interact with the world as seamlessly as humans do. A key aspect of human intelligence is the innate ability to recognize the functionalities offered by objects and environments —allowing us to effortlessly adapt to unstructured settings like homes. For AI agents to achieve similar generalization, they must learn how to interact with objects based on their intended purpose —a concept known as **affordance**. First introduced by psychologist James Gibson Gibson (2014), the concept of affordance has become an important topic for advancing AI and robot capabilities in our daily life. A plethora of studies have been conducted on affordances for visual recognition Hou et al. (2021); Hong et al. (2023), action prediction Roy & Fernando (2021); Chen et al. (2023), and functionality understanding Li et al. (2023a); Zhang et al. (2023); Kim et al. (2024). Understanding affordances through the lens of human-object interactions (HOIs) also offers a compelling approach for teaching AI agents. By observing how humans manipulate and interact with objects, we can extract rich cues about objects' functionality, thus enabling a broader set of interactions for AI agents.

However, prior work in HOI affordance learning has largely focused on *contact-based* affordances, which is a restrictive subset of all possible affordances. For instance, recent methods estimate contact scores from RGB images Bahl et al. (2023; 2022); Li et al. (2023a), 3D point clouds Chu et al. (2025); Yang et al. (2023), or human models Hassan et al. (2021), by relying on densely annotated human-object contact labels. This manual supervision is not only labor-intensive but also fails to generalize to novel objects and broader classes of interaction.

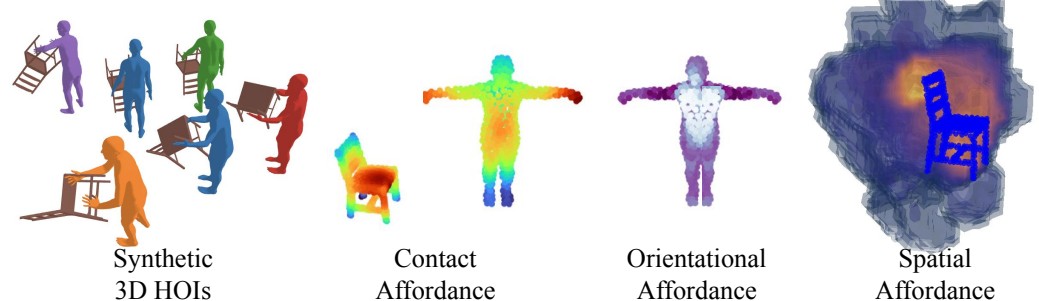

Figure 1: H2OFlow learns comprehensive affordances from synthetic 3D HOI data generated by 3D generative models using a novel representation. The learned affordance captures contact, orientational, and occupancy information based on input object point clouds.

We observe that human-object interactions (HOIs) involve 3D spatial relationships beyond simple contact. For example, human faces, torsos, and arms often maintain characteristic distances and orientations relative to objects, with natural variations across interactions. For instances, humans would grasp different tools with different hand configurations: a hammer is typically held at a specific distance from its head, with the wrist angled to allow effective striking, while a pen is gripped closer to the tip for finer control. A complete understanding of affordances in HOIs should incorporate these geometric patterns, including relative positioning and orientational tendencies.

A recent work by Kim et al. (2024) introduces the concept of *comprehensive affordance*, which captures these relationships probabilistically. Instead of binary contact labels, their method models a distribution over possible 3D spatial and orientational relations between every pair of object and human surface points. This approach generalizes affordance reasoning beyond contact, enabling finer-grained understanding of interaction geometry. As shown in Kim et al. (2024), learning comprehensive affordances in HOIs typically relies on synthetic RGB images uplifted to 3D using 2D-to-3D techniques. However, this approach requires intricate masking methods to achieve high-quality results, introducing multiple potential failure modes. Furthermore, the learned affordances often fail to generalize to novel real-world objects, and the dependency on well-defined watertight meshes for better-quality affordance computation severely limits real-world applicability.

To address these challenges, we leverage recent advances in 3D generative models for HOIs Li et al. (2024a); Diller & Dai (2024); Peng et al. (2023). Our key innovation is a pipeline that directly generates plausible 3D HOI samples using generative models, eliminating the need for error-prone 2D-to-3D uplifting. To ensure generalization to novel geometries, we subsample points from the generated data and employ dense diffused flows Eisner et al. (2022) —a technique proven effective for modeling multi-modality— to reconstruct 3D humans from the HOI samples. For comprehensive affordance learning, we introduce a novel probabilistic formulation operating directly on human-object point cloud pairs, circumventing the need for a watertight mesh.

This culminates in **Human-Object Flow** (**H2OFlow**), a framework for learning rich affordance knowledge in HOIs (Fig. 1). Our key contributions are:

1. A point-cloud-based affordance representation that efficiently captures both explicit contact and implicit non-contact interaction patterns in HOIs from raw point clouds inputs.
2. A synthetic data generation and learning pipeline, which leverages 3D generative models and dense diffused flows, that learns flexible affordances from synthetic 3D point clouds.
3. Extensive quantitative and qualitative experiments demonstrating the effectiveness and practical utility of the learned affordances on both synthetic datasets and real-world data.

## 2  RELATED WORK

**Affordance Learning.** First introduced in Gibson (2014), affordance learning has emerged as a critical capability for AI and robotic systems. Modern approaches focus on enhancing agents' ability for better visual recognition Hou et al. (2021); Hong et al. (2023), action prediction Roy & Fernando (2021); Chen et al. (2023), functionality understanding Li et al. (2023a); Eisner et al. (2022); Kim

et al. (2024), mimicking scene-conditioned human-object and hand-object interactions Bhatnagar et al. (2022); Lu et al. (2022); Nguyen et al. (2024); Jiang et al. (2022); Huang et al. (2022); Jiang et al. (2022); Petrov et al. (2023); Hassan et al. (2021); Zhang et al. (2022). With the advances of LLMs, more works have been proposed to explore open-vocabulary affordances in point clouds Nguyen et al. (2024); Chu et al. (2025). However, most works focus exclusively on *contact-based* affordances, neglecting crucial spatial and orientational aspects of interactions. Moreover, the requirement of manual labeling of the contact regions Do et al. (2018); Jian et al. (2023); Tripathi et al. (2023); Delitzas et al. (2024); Yang et al. (2023) is deemed cumbersome and restrictive when generalizing to the real world. More recently, Kim et al. (2024) proposed a comprehensive set of affordance representations that captures both contact and non-contact knowledge in HOIs without manual labels. While such comprehensive affordances work well in capturing both contact and spatial relations, they require to calculate the normal directions of each vertex and the inferred affordances have limited generalization to novel objects. We instead propose a novel set of affordance representations that operates on (partially observed) point cloud data, bypassing the need of watertight meshes, and generalizes to unseen objects via learned dense diffused flows.

**3D Flows in Visual Learning.** 3D flows have emerged as a powerful representation in visual learning, playing a key role in both policy learning Hu et al. (2017); Bahl et al. (2022; 2023) and object understanding Eisner et al. (2022); Xu et al. (2024); Cai et al. (2024). By capturing how points in 3D space move over time, 3D flows inherently encode affordances under external forces. For instance, predicting flow on articulated objects reveals how individual parts might move when interacted with by a human. While prior work has largely focused on learning 3D flows for rigid objects, we extend this intuition to the human body. Specifically, we propose to learn 3D flows that predict how each point on the human body moves when interacting with an object. Given the multi-modal and highly deformable nature of human-object interactions (HOIs), we leverage diffusion models Ho et al. (2020); Rombach et al. (2022); Ramesh et al. (2022); Nakayama et al. (2023); Peebles & Xie (2023) to learn these flows in a dense and expressive manner. We refer to this representation as *dense diffused flows*. As we show later, dense diffused flows generalize well to unseen objects and we are able to infer comprehensive affordance knowledge using such a representation.

**HOI Data Synthesis in 3D.** With the growing availability of paired scene-motion datasets Araújo et al. (2023); Hassan et al. (2019); Wang et al. (2022b); Zheng et al. (2022), a range of methods has been developed to synthesize human interactions in 3D environments Brahmbhatt et al. (2019a;b; 2020); Araújo et al. (2023); Hassan et al. (2021); Wang et al. (2022a); Taheri et al. (2020); Zhou et al. (2022); Ye et al. (2023). Another line of research leverages reinforcement learning to train scene-aware policies that generate navigation and interaction motions in static 3D scenes Xiao et al. (2023); Lee & Joo (2023). More recently, with the rise of large language models and the availability of paired human-object motion data Bhatnagar et al. (2022); Li et al. (2023b), several works have demonstrated the ability to predict human-object interactions (HOIs) from sparse waypoints or textual descriptions Li et al. (2024a); Diller & Dai (2024); Peng et al. (2023), enabling direct generation of 3D HOI data from language. In H2OFlow, we leverage the pre-trained model from Li et al. (2024a) to synthesize a diverse set of HOI sequences from text. These sequences are rich in affordance cues that go beyond mere contact information. We then subsample vertices from the resulting human-object meshes to generate point clouds for downstream learning of dense diffused flows. At inference time, our model requires only a partially observed object point cloud to infer affordances.

## 3 PROBLEM FORMULATION

We address the problem of learning comprehensive human-object interactions (HOIs) from point cloud data. Given a human point cloud $\boldsymbol{H} = \{\boldsymbol{h}_i\}_{i=1}^{N_H} \in \mathbb{R}^{N_H \times 3}$ and an object point cloud $\boldsymbol{O} = \{\boldsymbol{o}_j\}_{j=1}^{N_O} \in \mathbb{R}^{N_O \times 3}$, our goal is to infer a novel affordance representation that captures three key aspects of interaction: contact, orientation, and spatial configuration. Figure 2) provides an overview of H2OFlow.

We define an affordance score for each pair of human-object points $(i, j)$. The **contact affordance**, denoted as $C_{ij} \in \mathbb{R}$, reflects the likelihood of contact between human point $\boldsymbol{h}_i$ and object point $\boldsymbol{o}_j$, with higher values indicating actual contact. The **orientational affordance**, denoted as $R_{ij} \in \mathbb{R}$, captures the characteristic orientation patterns of human body parts relative to the object (*e.g.*, the

forearms' rotation relative to the table top is more uniform than the feet's in Figure 2). A higher $R_{ij}$ value indicates a consistent and meaningful orientation pattern observed during interactions. The **spatial affordance**, denoted as $S_{ij} \in \mathbb{R}^{H \times W \times L}$, over a voxel grid of size $H \times W \times L$, characterizes the spatial occupancy of human body parts around the object, assigning higher scores to regions frequently occupied during interactions in 3D space (*e.g.*, the orange region in the spatial affordance of Figure 2 tends to get occupied by human more than the purple region).

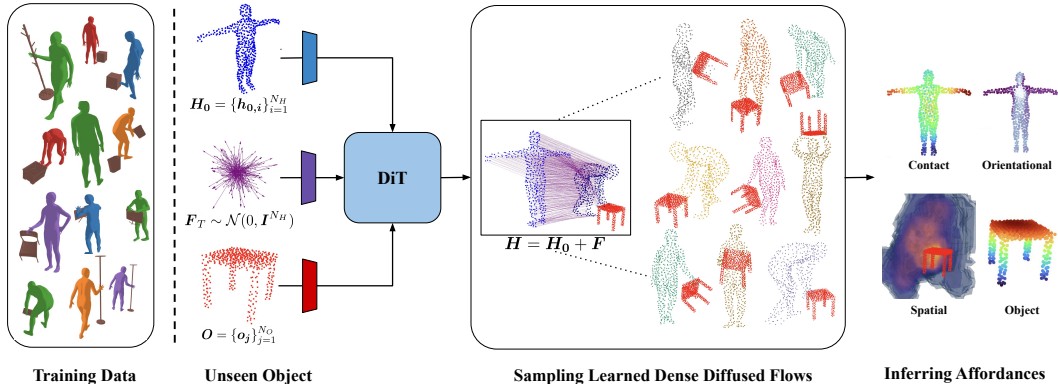

Figure 2: H2OFlow overview. We generate synthetic 3D HOI mesh samples, process the meshes into a point cloud and train DiT to learn a dense diffused flow distribution for human goal configuration prediction. Upon seeing an unseen object, H2OFlow samples learned dense flows to reconstruct goal humans. Using the flows and point clouds, we are able to infer comprehensive affordances. Note the "Object" affordance here is the transpose of the human contact affordance matrix.

## 4  METHOD

To learn affordance knowledge from point clouds in a generalizable manner, we propose **H2OFlow**, a framework that first synthesizes diverse human-object interaction (HOI) samples using a pre-trained 3D generative model as the *training data*. Then, we train a *diffusion model* that takes as input an object point cloud and predicts human interactions in the form of dense diffused flows Xu et al. (2024), a probabilistic representation that predicts per-point displacement on the human point cloud conditioned on the HOI. During *inference*, these flows are then used to comprehensively infer HOI affordances —contact, orientation, and spatial— directly from the object point cloud.

### 4.1  TRAINING DATA: SYNTHETIC HOI SAMPLES GENERATION

We employ a pretrained 3D generative model to generate diverse and realistic HOI mesh sequences. Given an initial object-human configuration and a language prompt, the pre-trained generative model generates temporally synchronized object and human motions. The outputs are long mesh sequences comprising varied and rich interaction dynamics across different object categories. Please refer to the Training Data illustration of Figure 2 for examples.

One might ask: *why can't we directly learn affordances from generative model outputs?* There are two main problems that hinder the generalizability of directly inferring affordances from synthetic HOIs Kim et al. (2024). First, such generative models are trained with object meshes, while inputs from raw-sensor data contain noisy point clouds, making such an approach incompatible with real-world data. Second, it is costly to generate and analyze 3D HOI meshes, creating a large computational and memory bottleneck. Thus, it is imperative for us to find a way that generalizes well to unseen point clouds for practicality, while maintaining a lower computational cost. We make use of *dense diffused flow*, a representation that lends itself well to point cloud learning.

### 4.2  AN INTERMEDIATE REPRESENTATION: DENSE FLOWS

To better generalize to unseen objects, H2OFlow reconstructs plausible human configurations from a given object point cloud $O$ using an *intermediate, point-based* representation, **dense flows** Zhang

et al. (2023); Xu et al. (2024); Cai et al. (2024), which can be applied to both rigid and deformable objects. Dense flows represent how each point transitions from its initial to its target configuration.

We assume the initial human pose is given by a standard 0-pose (T-pose) SMPL mesh[1]. From this mesh, we sample $N_H$ points $\{\pi(1), \pi(2), ..., \pi(N_H)\}$ to construct the initial human point cloud $\boldsymbol{H}_0$, where $\pi(\cdot)$ denotes the sampling operator. To obtain the goal human configuration from a synthesized HOI mesh, we sample the same $N_H$ points to create the goal human point cloud $\boldsymbol{H}$, ensuring one-to-one correspondence.

Using this setup, we compute the dense flow field $\boldsymbol{F} = \{\boldsymbol{f}_i\}_{i=1}^{N_H}$ as the per-point displacement between the goal and initial configurations of the human:

$$\boldsymbol{f}_i := \boldsymbol{h}_i - \boldsymbol{h}_{0,i}, \quad \forall i \in \{1, \ldots, N_H\}, \tag{1}$$

which can be compactly written as $\boldsymbol{F} := \boldsymbol{H} - \boldsymbol{H}_0$. As illustrated in the Learned Dense Diffused Flows in Figure 2, we can sample $\boldsymbol{F}$ conditioned on the object point cloud to reconstruct diverse goal human point clouds. We provide more detailed dense flows visualization in Figure 6 of Section C. In summary, given a generic 0-pose human point cloud $\boldsymbol{H}_0$ and an input object point cloud $\boldsymbol{O}$, H2OFlow predicts a dense flow field that displaces $\boldsymbol{H}_0$ into a realistic interaction configuration $\boldsymbol{H}$, effectively modeling the human-object interaction through spatial deformation[2].

### 4.3 LEARNING THE DENSE FLOWS REPRESENTATION

Human-object interactions (HOIs) in both real-world scenarios and synthesized samples exhibit strong multimodality. For instance, a human may contact an object using either the left or right hand, or interact with different regions of the same object. This diversity highlights the need for a *distributional* representation of dense flow that captures a continuous spectrum of plausible human configurations, rather than a single deterministic outcome.

To this end, we aim to learn a distribution over dense flows conditioned on an object point cloud: $g(\boldsymbol{H}_0, \boldsymbol{O}) = p_\theta(\boldsymbol{F} \mid \boldsymbol{O})$, where $\boldsymbol{H}_0$ is the initial human point cloud and $\boldsymbol{F}$ is the dense flow field. At inference time, we can sample a plausible dense flow $\boldsymbol{F} \sim p_\theta(\boldsymbol{F} \mid \boldsymbol{O})$ and reconstruct a goal human configuration via $\boldsymbol{H} = \boldsymbol{H}_0 + \boldsymbol{F}$.

To effectively model this complex distribution, we adopt diffusion models Ho et al. (2020); Sohl-Dickstein et al. (2015); Peebles & Xie (2023), which learn data distributions through iterative forward noising and reverse denoising processes. By applying this framework to dense flow prediction, we introduce the concept of *dense diffused flow*, enabling our model to generate diverse and plausible human poses in interaction with a given object.

**Diffusion Process.** Given a synthetic HOI sample point cloud pair $(\boldsymbol{H}, \boldsymbol{O})$ and a canonical 0-pose human point cloud $\boldsymbol{H}_0$, we train a diffusion model to learn the distribution over dense flows $\boldsymbol{F}$. The ground-truth dense flow is defined as the per-point displacement between the goal and initial configurations:

$$\boldsymbol{F}_{GT} = \boldsymbol{H} - \boldsymbol{H}_0. \tag{2}$$

Following standard diffusion modeling practices Ho et al. (2020); Song et al. (2020), we construct a noisy version of the clean dense flow $\boldsymbol{F}_0 := \boldsymbol{F}_{GT}$ by sampling at time step $t \sim \{1, \ldots, T\}$: $\boldsymbol{F}_t = \sqrt{\bar{\alpha}_t}\boldsymbol{F}_0 + \sqrt{1 - \bar{\alpha}_t}\boldsymbol{\epsilon}$, where $\boldsymbol{\epsilon} \sim \mathcal{N}(0, \boldsymbol{I})$ is Gaussian noise, and $\bar{\alpha}_t$ is the cumulative product of noise scheduling parameters $\beta_t$. The forward process adds Gaussian noise progressively over time steps, while the reverse process learns to denoise and recover the original $\boldsymbol{F}_0$. We parameterize the reverse process as: $p_\theta(\boldsymbol{F}_{t-1} \mid \boldsymbol{F}_t) = \mathcal{N}(\boldsymbol{F}_{t-1}; \boldsymbol{\mu}_\theta(\boldsymbol{F}_t), \boldsymbol{\Sigma}_\theta(\boldsymbol{F}_t))$, and supervise the model using the hybrid loss from Nichol & Dhariwal (2021) that combines the noise loss with a new cumulative KL- loss using the derived $\boldsymbol{\Sigma}_\theta(\boldsymbol{F}_t)$.

During inference, given an object point cloud $\boldsymbol{O}$ and a generic 0-pose human point cloud $\boldsymbol{H}_0$, we initialize the dense diffused flows as Gaussian noise: $\boldsymbol{F}_T \sim \mathcal{N}(0, \boldsymbol{I})$. The dense diffused flows are iteratively denoised via the reverse process. The final denoised flows $\boldsymbol{F}_0$ are then used to transform the points of $\boldsymbol{H}$ into a predicted interaction configuration $\boldsymbol{H}_0 + \boldsymbol{F}_0$ with respect to the object $\boldsymbol{O}$.

---

[1]Please refer to Appendix Section B for details on placing the 0-pose human relative to the object.

[2]Dense flows representation is the fundamental reason for H2OFlow's generalizability, and we discuss this design and advantages over prior works in more details in Appendix Section G.

**Dense Diffused Flows from Diffusion Transformer.** Diffusion Transformers (DiT) Peebles & Xie (2023) have demonstrated strong capability in modeling multi-modal point cloud distributions for deformable objects Cai et al. (2024). We adopt DiT as the backbone for predicting dense diffused flows. At each diffusion timestep, the model takes as input the noised flow $\boldsymbol{F}_t$, the human point cloud $\boldsymbol{H}$, the object point cloud $\boldsymbol{O}$, and the timestep $t$. Using MLP encoders with shared weights, we extract per-point features from each input: dense flow features $f^{\boldsymbol{F}}$ from $\boldsymbol{F}_t$, human features $f^{\boldsymbol{H}}$ from $\boldsymbol{H}$, and object features $f^{\boldsymbol{O}}$ from $\boldsymbol{O}$. The dense flow and human features are concatenated to form joint features $f^{\boldsymbol{FH}}$, which serve as the input to the DiT model, conditioned on the object features $f^{\boldsymbol{O}}$. Within each DiT block, self-attention is first applied to the joint human-flow features $f^{\boldsymbol{FH}}$ to enable local reasoning across the human point cloud and coordinate flow predictions. Then, cross-attention is applied between $f^{\boldsymbol{FH}}$ and the object features $f^{\boldsymbol{O}}$ to capture global human-object interaction patterns. This process is repeated across $N$ DiT blocks, after which the network outputs the predicted noise $\boldsymbol{\epsilon}_\theta$ and the interpolation vector $\boldsymbol{v}_\theta$. We explain the training objective (hybrid loss) and details in Section F.

## 4.4 TEST TIME: COMPREHENSIVE AFFORDANCE INFERENCE

During inference, with the learned diffusion model, we can sample flows conditioned on an object point cloud, resulting in a distribution over possible human goal configurations. Given an initial human point cloud $\boldsymbol{H}_0$ and a sampled flow $\boldsymbol{F} \sim p_\theta(\boldsymbol{F} \mid \boldsymbol{O})$, the goal human is given by $\boldsymbol{H} = \boldsymbol{H}_0 + \boldsymbol{F}$ and each point of the sampled predicted human $\boldsymbol{h}_i = \boldsymbol{h}_{0,i} + \boldsymbol{f}_i$. For each predicted human point $\boldsymbol{h}_i$, we define a conditional probability distribution with respect to each object point $\boldsymbol{o}_j$:

$$\mathcal{P}_{ij} := p(\boldsymbol{h}_i \mid \boldsymbol{o}_j) \tag{3}$$

Thus, $\mathcal{P}_{ij}$ defines the possible human points locations in diverse HOI samples. In practice, this distribution is defined over a large set of generated HOI samples. Our three affordance types — *contact, orientational, and spatial*— are then defined over this pairwise distribution $\mathcal{P}_{ij}$, resulting in a per-point-pair evaluation of affordance.

**Contact Affordance.** We define the contact affordance score $c_{ij}$ between human point $\boldsymbol{h}_i$ and object point $\boldsymbol{o}_j$ as:

$$c_{ij} = \mathbb{E}_{\boldsymbol{h}_i \sim \mathcal{P}_{ij}} \left[ w_{ij} \cdot \frac{\exp\left(-\|\boldsymbol{d}_{ij}\|\right)}{\tau} \right], \tag{4}$$

where $\boldsymbol{d}_{ij} = \boldsymbol{h}_i - \boldsymbol{o}_j$ denotes the per-pair displacement between human point and object point, $w_{ij}$ denotes the cross-attention weight between $\boldsymbol{h}_i$ and $\boldsymbol{o}_j$ from the DiT model, and $\tau$ is a temperature hyperparameter that controls sensitivity to distance.

Intuitively, the contact affordance score $c_{ij}$ is higher when the human and object points are likely to be spatially close during HOI. The inclusion of the cross-attention weight $w_{ij}$ further enhances contact prediction by leveraging semantic alignment from the DiT model, especially in cases where contact is not perfectly captured in the sampled HOI configurations.

**Orientational Affordance.** Following the intuition from prior work Kim et al. (2024), we aim to capture the consistency and pattern of human body part orientations relative to object geometry using an entropy-based formulation. The key idea is that a lower entropy in the orientation distribution implies a stronger, more consistent orientational pattern during interaction, indicating a high orientational affordance. However, unlike Kim et al. (2024), which computes surface normals to measure orientation–often computationally expensive and unstable under noisy meshes–we leverage the predicted *dense diffused flows* directly as a proxy for directional motion. Specifically, for each human-object point pair $(i, j)$, we compute a relative orientation vector using the cross product between the displacement vector $\boldsymbol{d}_{ij}$ (from human point $\boldsymbol{h}_i$ to object point $\boldsymbol{o}_j$) and the diffused flow vector $\boldsymbol{f}_i$:

$$\boldsymbol{x}_{ij} = \frac{\boldsymbol{d}_{ij} \times \boldsymbol{f}_i}{\|\boldsymbol{d}_{ij} \times \boldsymbol{f}_i\|}. \tag{5}$$

The cross-product $\boldsymbol{x}_{ij}$, intuitively, represents the relative displacement direction between the human and the object given the human dense flow direction, efficiently grounding the per-pair information on the overall flow direction. To evaluate the distribution of these orientation vectors, we discretize

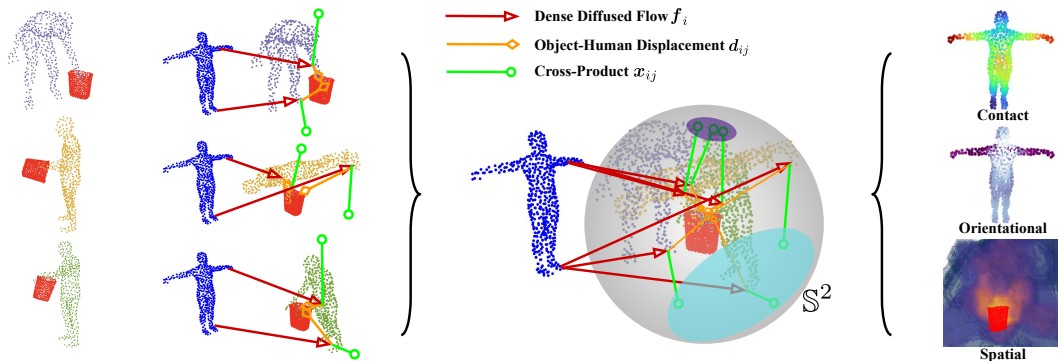

Figure 3: Visual illustration of affordance inference. Given predicted human point clouds, contact affordance assigns high scores to human-object point pairs that are close. Orientational affordances give higher scores to point pairs that yield more uniform cross-product directions (*i.e.*, hand points) and vice versa (*i.e.*, foot points). The spatial affordances output higher scores to regions surrounding the object that are often occupied by human parts. A video of the figure is available at this website.

the unit sphere $\mathbb{S}^2$ into $n_b$ bins with representative directions $\{\boldsymbol{n}_1, \ldots, \boldsymbol{n}_{n_b}\}$. The discrete probability of $\boldsymbol{x}_{ij}$ falling into bin $n$ is computed using a Gaussian kernel:

$$p_{\boldsymbol{x},ij}(n) \propto \exp\left(-\frac{\|\boldsymbol{x}_{ij} - \boldsymbol{n}_n\|^2}{2\sigma^2}\right), \quad n = 1, \ldots, n_b, \tag{6}$$

where $\sigma$ is a hyperparameter. This defines a distribution over orientation bins on the sphere. We then compute the negated Shannon entropy of this distribution:

$$\mathcal{H}_{ij} = \mathbb{E}_{n \sim \mathbb{S}^2}\left[\log p_{\boldsymbol{x},ij}(n)\right], \tag{7}$$

which becomes higher when orientations concentrate around specific directions.

Finally, we define the orientational affordance score $R_{ij}$ as the expectation of this negated entropy over the distribution of possible human configurations:

$$R_{ij} = \mathbb{E}_{\boldsymbol{h}_i \sim \mathcal{P}_{ij}}\left[w_{ij} \cdot \frac{\mathcal{H}_{ij}}{\tau}\right], \tag{8}$$

where $w_{ij}$ is the cross-attention weight from the DiT model and $\tau$ is a temperature hyperparameter.

Since a uniform distribution has high entropy, while structured behavior has low entropy, a low $R_{ij}$ indicates that the orientation distribution $p_{\boldsymbol{x},ij}(n)$ is nearly uniformly random —*i.e.*, no dominant pattern exists— whereas a high $R_{ij}$ reflects consistent and structured orientational behavior in human-object interactions[3].

**Spatial Affordance.** Lastly, we aim to capture the 3D spatial occupancy pattern of human surface points with respect to object geometry, following ideas from prior work Han & Joo (2023); Kim et al. (2024). This affordance measures the likelihood that a specific region in space is occupied by a part of the human body during interaction with the object.

We define a voxel grid $\boldsymbol{G} \in \mathbb{R}^{H \times W \times L}$, covering the spatial region around the object. For each voxel $g \in \boldsymbol{G}$, we introduce an indicator function $\delta_{ij}$ that equals 1 if the voxel $g$ contains the human point $\boldsymbol{h}_i$, and 0 otherwise. The spatial affordance score is then defined as the expected occupancy of voxel $g$ by point $\boldsymbol{h}_i$, conditioned on the interaction with object point $\boldsymbol{o}_j$:

$$S_{ij} = \mathbb{E}_{\boldsymbol{h}_i \sim \mathcal{P}_{ij}}[\delta_{ij}] \tag{9}$$

This formulation results in a discrete occupancy map over the voxel grid, which can be further analyzed as a spatial probability distribution. Learning spatial affordance helps us understand the typical spatial arrangement or positioning of the human body relative to the object during interaction.

---

[3]We propose advanced use cases of orientational affordance in Section N.2.

| Baseline | SIM-H ↑ | SIM-O ↑ | MAE-H ↓ | MAE-O ↓ | Precision@K ↑ | MSE ↓ |
|---|---|---|---|---|---|---|
| COMA | $41.3 \pm 2.2\%$ | $56.9 \pm 1.4\%$ | $0.22 \pm 0.07$ | $0.14 \pm 0.03$ | $42.9 \pm 7.2\%$ | $0.14 \pm 0.06$ |
| COMA-Recon | $20.7 \pm 4.1\%$ | $31.8 \pm 1.9\%$ | $0.62 \pm 0.11$ | $0.51 \pm 0.05$ | $9.1 \pm 2.4\%$ | $0.66 \pm 0.12$ |
| H2OSMPL | $57.3 \pm 2.1\%$ | $68.0 \pm 3.3\%$ | $0.21 \pm 0.03$ | $0.15 \pm 0.03$ | $53.6 \pm 1.9\%$ | $0.14 \pm 0.01$ |
| H2OFlow-NoAttn | $67.3 \pm 1.6\%$ | $76.4 \pm 2.4\%$ | $0.15 \pm 0.02$ | $0.09 \pm 0.01$ | $69.2 \pm 1.1\%$ | $0.12 \pm 0.01$ |
| **H2OFlow** | $\mathbf{72.3 \pm 1.3\%}$ | $\mathbf{81.0 \pm 2.4\%}$ | $\mathbf{0.11 \pm 0.03}$ | $\mathbf{0.07 \pm 0.01}$ | $\mathbf{75.6 \pm 3.1\%}$ | $\mathbf{0.12 \pm 0.01}$ |

Table 1: Quantitative comparisons with various baselines on OMOMO dataset. Note that -H and -O represent human and object contact results.

In practice, this representation avoids reliance on high-quality surface meshes and is highly efficient: operations are parallelizable on GPUs, and memory usage is minimized by sampling only a small subset of points from both the human and object point clouds.

## 5 EXPERIMENTS

We present both quantitative and qualitative results to evaluate H2OFlow. We use a pretrained CHOIS Li et al. (2024a) as the 3D generative model backbone to generate diverse HOIs. During training, we apply random perturbation and occlusion to the objects point cloud to achieve real-world robustness. We compare against baseline methods in terms of affordance learning quality, memory efficiency, and runtime performance. For the qualitative evaluation, we demonstrate how H2OFlow surpasses traditional contact-based affordances via distributions over orientational and spatial information across a diverse range of object categories.

### 5.1 QUANTITATIVE RESULTS

**Baselines.** We compare H2OFlow against **COMA** Kim et al. (2024) using objects from the OMOMO test set Li et al. (2023b). Since COMA requires 2D object images to generate inpainted HOI samples, we render each OMOMO object from 50 camera views. To ensure a fair comparison, we also reconstruct object meshes from H2OFlow's point cloud inputs and render them from the same views as input to COMA —this serves as the **COMA-Recon** baseline. We include a variant of our method, **H2OSMPL**, where we learn a direct SMPL predictor using diffusion conditioned on the object input. Additionally, we include a variant of our method, **H2OFlow-NoAttn**, which removes the cross-attention mechanism used for aggregating affordance scores. All methods generate 50 HOI samples per object for evaluation.

**Metrics.** For *contact* affordance, we compute the similarity (**SIM**) Swain & Ballard (1991) and mean absolute error (**MAE**) between the normalized predicted and ground-truth contact distributions. For *orientational* affordance, we rank human vertices by the average entropy of their relative orientations in ground-truth HOIs, and compare these to rankings based on the predicted orientational scores. We report **Precision@K** by measuring the overlap between the top-K ranked sets. For *spatial* affordance, we calculate the **mean squared error (MSE)** between predicted and ground-truth voxel occupancy grids.

**Results.** As seen in Table 1, for all metrics, H2OFlow outperforms other baselines by a very noticeable margin. We note that COMA's performance breaks when the input 2D rendered mesh images are reconstructed from point clouds. Moreover, learning dense diffused flows results in better performance than learning SMPL parameters directly. We analyze why this is the case in Appendix Section L. H2OFlow performs better with attention weights in contact and orientational affordances aggregation. We provide results on the BEHAVE dataset and contact-only baselines in Appendix Section I and Section J. Note that COMA, to our knowledge, is the only prior work addressing different types of affordances. We test H2OFlow's robustness against occlusion in Appendix Section K.

**Memory and Runtime Comparisons.** Experiments suggest that H2OFlow utilizes significantly less memory and runs faster than COMA. This is expected in that H2OFlow operates on sparse point clouds. We document quantitative comparisons in Appendix Section M.

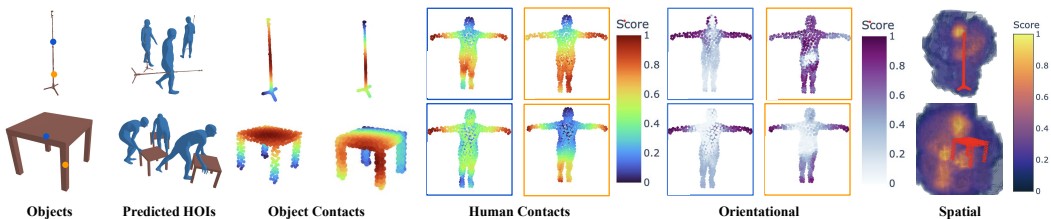

Figure 4: Visualizations of affordances inferred from flows prediction with color maps. H2OFlow infers diverse affordance distributions from predicted HOI samples on unseen objects.

## 5.2 QUALITATIVE RESULTS

**Learned Affordances Visualizations.** We showcase sample inferred affordances in Figure 4. The qualitative results are also run on the test objects of the OMOMO dataset, unseen during the training of H2OFlow. For each object, we pick two points on the object that give us interesting interaction information. Both contact affordance and orientation affordance reflect diverse, multimodal distributions from the predicted HOI samples. Depending on the points on the object, human contact affordances reflect different heatmaps, and different parts of the human also exhibit different orientational tendencies. In Figure 8, we provide more examples. Specifically, in the monitor example of Figure 8, the selected bottom (orange) point makes more contact with the side of the body while the center (blue) point makes frequent contact with the whole torso, which reflects real-world contact tendency when moving a monitor. For the tripod in Figure 8, human legs tend to exhibit a more uniform orientation relative to the bottom of the tripod (orange) than their hands, while the orientation of the hands relative to the top part of the tripod (blue) is more uniform. For spatial affordance, we can see the high-level human occupancy around the object during HOIs: the high-probability regions are more frequently occupied by the human body parts, consistent with real-world interactions (in some cases, full human silhouettes are observed).

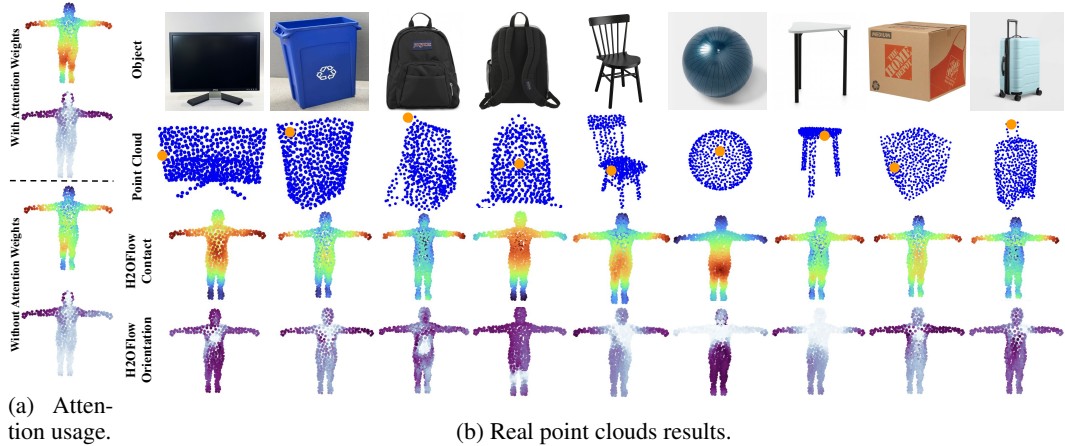

(a) Attention usage.

(b) Real point clouds results.

Figure 5: (a) Ablations on cross-attention weights and (b) results on real-world point clouds. Objects shown are: monitor, trashcan, backpack handle & panel, chair, yoga ball, table, box, and suitcase.

**Cross-Attention Weights Ablation.** We ablate the effect of incorporating cross-attention weights into the computation of affordance scores, as shown in Figure 5a. With cross-attention weights, both the contact and orientational affordances exhibit greater symmetry compared to the variant without attention. This is particularly valuable in low-sample scenarios, where sampling only a few instances from the diffusion model may result in limited diversity and fail to fully capture the underlying multimodal distribution. Cross-attention weights mitigate this issue by learned geometric associations between human and object. Even when the sampled outputs are sparse, the attention weights act as a corrective signal, producing plausible and semantically aligned affordance estimations.

**Affordance Types Ablation.** We are interested in studying the contribution of *orientational* and *spatial* affordances beyond *contact*. We design two downstream HOI inference tasks and vary only which affordance terms are computed and used for downstream scoring. We evaluate downstream HOI inference tasks on unseen objects and test on the following variants of H2OFlow affordances output: (1) **C**: use $c_{ij}$ only, (2) **C+O**: use $c_{ij}$ and $R_{ij}$, (3) **C+S**: use $c_{ij}$ and $S_{ij}$, (4) Shuffled: keep $c_{ij}$ but *randomly permute* $R_{ij}$ across human indices or $S_{ij}$ across voxels per object, and (5) **C+O+S**: use all three affordances (H2OFlow default). To combine terms for downstream scoring we use a normalized linear fusion $\phi_{ij} = \lambda_c \widehat{c}_{ij} + \lambda_o \widehat{R}_{ij} + \lambda_s \widehat{S}_{ij}$. We design two downstream HOI inference tasks: (1) **HOI Region Retrieval**: Given an object query point $o_j$, rank human points by $\phi_{ij}$; compute mAP@{1,5,10} against GT contact points, and (2) **Pose Selection**: Given sampled HOI hypotheses per object, select $\arg\max_k \sum_{i,j} \phi_{ij}^{(k)}$. Report Top-$k$ accuracy vs. GT pose clusters, *collision rate* with object, and *contact leakage* that measures contacts on implausible parts.

| Variant | mAP@1↑ | mAP@5↑ | mAP@10↑ | Top-5 Acc.↑ | Collision↓ | Leakage↓ |
|---|---|---|---|---|---|---|
| **C** | $0.52 \pm 0.03$ | $0.60 \pm 0.02$ | $0.66 \pm 0.02$ | $0.41 \pm 0.04$ | $0.31 \pm 0.02$ | $0.27 \pm 0.03$ |
| **C+O** | $0.57 \pm 0.03$ | $0.63 \pm 0.02$ | $0.69 \pm 0.02$ | $0.55 \pm 0.03$ | $0.26 \pm 0.02$ | $0.22 \pm 0.02$ |
| **C+S** | $0.56 \pm 0.03$ | $0.62 \pm 0.03$ | $0.68 \pm 0.02$ | $0.51 \pm 0.04$ | $0.21 \pm 0.01$ | $0.20 \pm 0.02$ |
| **C+O+S** | $\mathbf{0.63 \pm 0.02}$ | $\mathbf{0.69 \pm 0.02}$ | $\mathbf{0.76 \pm 0.03}$ | $\mathbf{0.64 \pm 0.03}$ | $\mathbf{0.18 \pm 0.01}$ | $\mathbf{0.17 \pm 0.01}$ |
| Shuffled | $0.54 \pm 0.03$ | $0.61 \pm 0.02$ | $0.66 \pm 0.02$ | $0.44 \pm 0.03$ | $0.29 \pm 0.02$ | $0.26 \pm 0.02$ |

Table 2: Downstream HOI inference results. Left: Region Retrieval (mAP@{1,5,10}); Right: Pose Selection (Top-5 accuracy, collision rate ↓, and contact leakage ↓).

We record the results in Table 2. As results suggest, against **C**, **C+O** and **C+S** significantly improve the metrics, and **C+O+S** yields further gains. Shuffled controls eliminate these improvements, confirming that structured orientational and spatial affordances indeed improve performance for affordance learning, not merely because of additional feature capacity.

**Unseen Real-World Objects.** We evaluate H2OFlow on real-world point clouds captured using a cheap depth camera on an iPhone, collected by the RealityKit and subsampled via FPS Qi et al. (2017), in Figure 5b. Due to training-time perturbation and occlusion, H2OFlow does not require full object scan and is robust to real-world occlusions (*e.g.*bottom). H2OFlow produces highly plausible affordance scores on these real inputs, effectively capturing meaningful interaction patterns —particularly the orientational tendencies around the head region. For example, we observe that clean, multi-modal affordances are inferred in the backpack examples (different parts). While the objects were unseen during training, H2OFlow learns local geometric cues via dense diffused flows instead of memorizing global mesh templates. Thus, the output affordances are semantically meaningful and consistent with the actual usage of the interacted parts on the objects. In contrast, as the full comparison shows in Figure 7, COMA relies on 2D renderings from clean meshes, and thus struggles with noisy, reconstructed meshes derived from point clouds. This limitation severely degrades COMA's performance, resulting in oversimplified and unimodal affordance score maps.

# 6 Conclusion

We introduced **H2OFlow**, a novel framework for learning comprehensive 3D affordances from synthetic data using dense diffused flows. H2OFlow demonstrates strong generalization to unseen objects and is capable of capturing diverse contact, orientational, and spatial relationships underlying human-object interactions. Looking forward, we aim to extend this framework to support more fine-grained interaction tasks and downstream applications such as robot policy learning. In particular, incorporating more diverse interaction data and exploring robot-human affordance correspondence will be key directions for future research.

# 7 Ethics Statement

We take ethics very seriously and our research conforms to the ICLR Code of Ethics. Affordance learning is a well-established research area, and this paper inherits all the impacts of the research area, including potential for dual use of the technology in both civilian and military applications.

We believe that the work does not impose a high risk for misuse. Furthermore, the paper does not involve crowdsourcing or research with human subjects.

## 8 REPRODUCIBILITY STATEMENT

Our paper makes use of publicly available open-source datasets, ensuring that the data required for reproducing our results is accessible to all researchers. We have thoroughly documented all aspects of our model's training, including the architecture, hyperparameters, optimizer settings, learning rate schedules, and any other implementation details for achieving the reported results. Additionally, we specify the hardware and software configurations used for our experiments to facilitate replication. We anticipate that it should not be challenging for other researchers to reproduce the results and findings presented in this paper.

## ACKNOWLEDGEMENTS

This work was partially funded by ONR RAPID Program and by Carlone's NSF CAREER Award.

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

## A    PROMPTING THE 3D GENERATIVE MODEL

To prompt the 3D generative model CHOIS Li et al. (2024a), we follow standard practices documented in Li et al. (2024a), where the model first randomly generates a series of waypoints for the human to follow and we prompt HOI generation using suggested prompts from Li et al. (2024a). We list some examples below in Table 3.

| Raw Prompt | Normalized Prompt |
|---|---|
| Facing the back of the chair, lift the chair and then place the chair onto the floor. | Facing the back of the chair, lift the chair, move the chair, and then place the chair on the floor. |
| Lift and move the chair. | Lift the chair, move the chair, and put down the chair. |
| Grab the top of the chair, swing the chair. | Grab the top of the chair, swing the chair, and put down the chair. |
| Lift the chair over your head, walk and place the chair onto the floor. | Lift the chair over your head, walk and then place the chair on the floor. |
| Put your hand on the back of the chair at the top.  Pull on it to move it across the floor. | Put your hand on the back of the chair, pull the chair, and set it back down. |
| Lift the chair, rotate the chair and set it back down. | Lift the chair, rotate the chair, and set it back down. |
| Use the foot to scoot the chair to change its orientation. | Use the foot to scoot the chair to change its orientation. |
| Push the chair, then turn yourself around so you can then drag the chair behind you. | Push the chair, release the hands, then drag the chair, and set it back down. |
| Hold the chair and turn it around to face a diffferent orientation. | Hold the chair and turn it around to face a different orientation. |
| Grab one of the chair's legs and tilt it at an angle. | Grab the chair's legs, tilt the chair. |
| Kick the chair across the room. | Kick the chair, and set it back down. |
| Lift the chair, flip it upside down and place it on top of the table. | Lift the chair, flip it upside down and place it on top of the table. |
| Move the chair upside down from the table to the floor. | Move the chair upside down from the table to the floor. |
| Lift the chair, flip it upside down and place it on top of the table.  And then move the chair upside down from the table to the floor. | Lift the chair, flip it upside down and place it on top of the table.  And then move the chair upside down from the table to the floor. |
| Lift and move the chair.  Then kick the chair to move. | Lift the chair, move the chair, then kick the chair to move, and set it back down. |

Table 3: Examples of raw and normalized action prompts used in CHOIS.

## B    ZERO-POSE HUMAN CONFIGURATION

We center the zero-pose human and the object into the same canonical frame.  Specifically, we subtract the centroid of the object point cloud from both the object and the human point cloud. To make sure the model is rotation-equivariant, we apply random perturbations to the object point cloud during training data generation.  In this way, the zero-shape human will be agnostic to the object location and rotation during inference. Note that in Figure 6, we move the human point cloud to the side for better visibility. In reality, the object and zero-pose human will overlap.

## C  DENSE FLOWS GROUND TRUTH

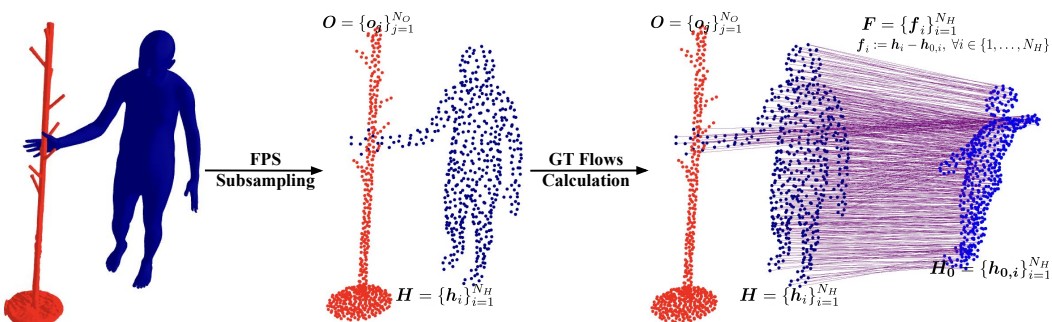

Figure 6: Dense flow training data generation visualization. Given a pair of HOI mesh generated from CHOIS, we first subsample the mesh vertices into point clouds using furthest point sampling (FPS) Qi et al. (2017). We then calculate the ground truth dense flows using Equation (1).

As shown in Figure 6, the ground-truth dense flows are calculated as the per-point displacement from a zero-pose SMPL model to the ground-truth HOI sample human, both of which are subsampled using the same set of indices.

## D  DATASET DETAILS

H2OFlow trains on OMOMO objects Li et al. (2023b), where the training set comprises 12 object categories while the testing dataset has 5 object categories. For each object in the training dataset, we generate 100 HOI sequences, where each sequence has 200 frames.

## E  HYPERPARAMETERS AND TRAINING DETAILS

We document the choice of hyperparmeters used in H2OFlow. For training the diffusion model, we use a learning rate of 1e-4, a weight decay of 1e-5, a batch size of 32, and a training epochs number of 20,000. Both human and object point clouds are downsampled to 512 points via FPS. During training, we apply random rotation to the objects and random occlusion as augmentation in order to ensure robustness to real-world variability. The model is trained with the AdamW optimizer, and the total number of steps is set to $T = 100$ in the diffusion process. Following Cai et al. (2024); Nichol & Dhariwal (2021), we use 128 as the hidden size per DiT block. We have 4 heads per block and 5 blocks in total.

In the inference of comprehensive affordances, we use a temperature hyperparameter of $\tau = 20$ for contact affordance $c_{ij}$. For orientational affordance, we use a variance of $\sigma^2 = 1$ and a temperature hyperparameter of $\tau = 10$.

During training and testing, we center the object coordinates and produce ground-truth in object frame (*i.e.*, the object is always upright).

## F  DIFFUSION MODEL DETAILS

We parameterize the reverse process as: $p_\theta(\boldsymbol{F}_{t-1} \mid \boldsymbol{F}_t) = \mathcal{N}(\boldsymbol{F}_{t-1}; \boldsymbol{\mu}_\theta(\boldsymbol{F}_t), \boldsymbol{\Sigma}_\theta(\boldsymbol{F}_t))$, where the mean $\boldsymbol{\mu}_\theta$ and variance $\boldsymbol{\Sigma}_\theta$ are derived from a predicted noise term $\boldsymbol{\epsilon}_\theta(\boldsymbol{F}_t, \boldsymbol{H}_0, \boldsymbol{O}, t)$ and an *interpolation vector* $\boldsymbol{v}_\theta(\boldsymbol{F}_t, \boldsymbol{H}_0, \boldsymbol{O}, t)$. Following Nichol & Dhariwal (2021), the interpolation vector contains one value per dimension and is used to parameterize the covariance: $\boldsymbol{\Sigma}_\theta(\boldsymbol{F}_t) = \exp\left(\boldsymbol{v}_\theta \log \beta_t + (1 - \boldsymbol{v}) \log \frac{1 - \bar{\alpha}_{t-1}}{1 - \bar{\alpha}} \beta_t\right)$. We supervise the model using the hybrid loss from Nichol & Dhariwal (2021) that combines the regular noise loss with a new cumulative KL-divergence loss using the derived $\boldsymbol{\Sigma}_\theta(\boldsymbol{F}_t)$.

## G    COMPARISON WITH COMA

### G.1    METHODOLOGY

We emphasize that H2OFlow is fundamentally different from COMA as H2OFlow introduces four technical advances over COMA.

First, more generalizability and flexibility. Most fundamentally, COMA directly uses reconstructed 3D inputs and has no learned components in their pipeline. While COMA lays out comprehensive affordances in a nice way, the lack of learning-based methods lacks its generalizability and flexibility when it comes to unseen objects (as we noted in the quantitative results of the paper).

Second, a point-cloud-based affordance learning paradigm with dense diffused flows as an effective intermediate representation. In COMA, affordances are inferred from reconstructed meshes instead of learned flow representations, which lack the generalizability to real-world visual inputs. More-over, COMA's per-vertex-pair affordance calculation between meshes consumes a lot more memory and time than H2OFlow's sparser per-point-pair formulation. In H2OFlow, with flows, no water-tight meshes or surface normals are needed, which tend to be noisy in real-worldscenarios . This is supported by results in Table 1 and Table 4, where COMA struggles to generalize to unseen objects and reconstructed meshes from noisy point clouds while H2OFlow performs well in both cases.

Third, diffusion-based multi-modal dense-flow predictor based on per-point encoding. This learning paradigm handles intrinsic ambiguity due to multi-modality and also learns to reason about geometric information on different regions (local information) of the object-human interaction. With dense diffused flows, H2OFlow's pipeline provides a new method for modeling human pose with a more flexible representation. At the same time, this representation sidesteps the need of normal vectors from meshes (Equation (5), Equation (6)), which are costly to compute for real-world applications, while achieving better results.

Lastly, cross-attention aggregation & partial-scan robustness. We improve the affordance aggregation via learned cross-attention weights (see ablations in Table 1). During training, we apply random rotation to the objects and random occlusion as augmentation in order to ensure robustness to real-world variability, which ensures the robustness to occlusion in the real world, as opposed to COMA that requires a clean mesh of the object.

### G.2    REAL-WORLD RESULTS

In Figure 7, we show comparisons with COMA on real-world unseen objects. COMA relies on 2D renderings from clean meshes, and thus struggles with noisy, reconstructed meshes derived from point clouds. This limitation severely degrades COMA's performance, resulting in oversimplified and unimodal affordance score maps.

## H    MORE QUALITATIVE RESULTS ON OMOMO DATASET

In Figure 8, we provide more examples. Specifically, in the monitor example, the selected bottom (orange) point makes more contact with the side of the body while the center (blue) point makes frequent contact with the whole torso, which reflects real-world contact tendency when moving a monitor. For the tripod, human legs tend to exhibit a more uniform orientation relative to the bottom of the tripod (orange) than their hands, while the orientation of the hands relative to the top part of the tripod (blue) is more uniform. For spatial affordance, we can see the high-level human occupancy around the object during HOIs: the high-probability regions are more frequently occupied by the human body parts, consistent with real-world interactions (in some cases, full human silhouettes are observed).

## I    RESULTS ON BEHAVE DATASET

We also test our method on BEHAVE dataset. To test the generalizability of H2OFlow, we provide a baseline version by testing on BEHAVE objects directly without fine-tuning (**H2OFlow-NoFT**). We also have another baseline that was finetuned on the BEHAVE objects (**H2OFlow-FT**). The results

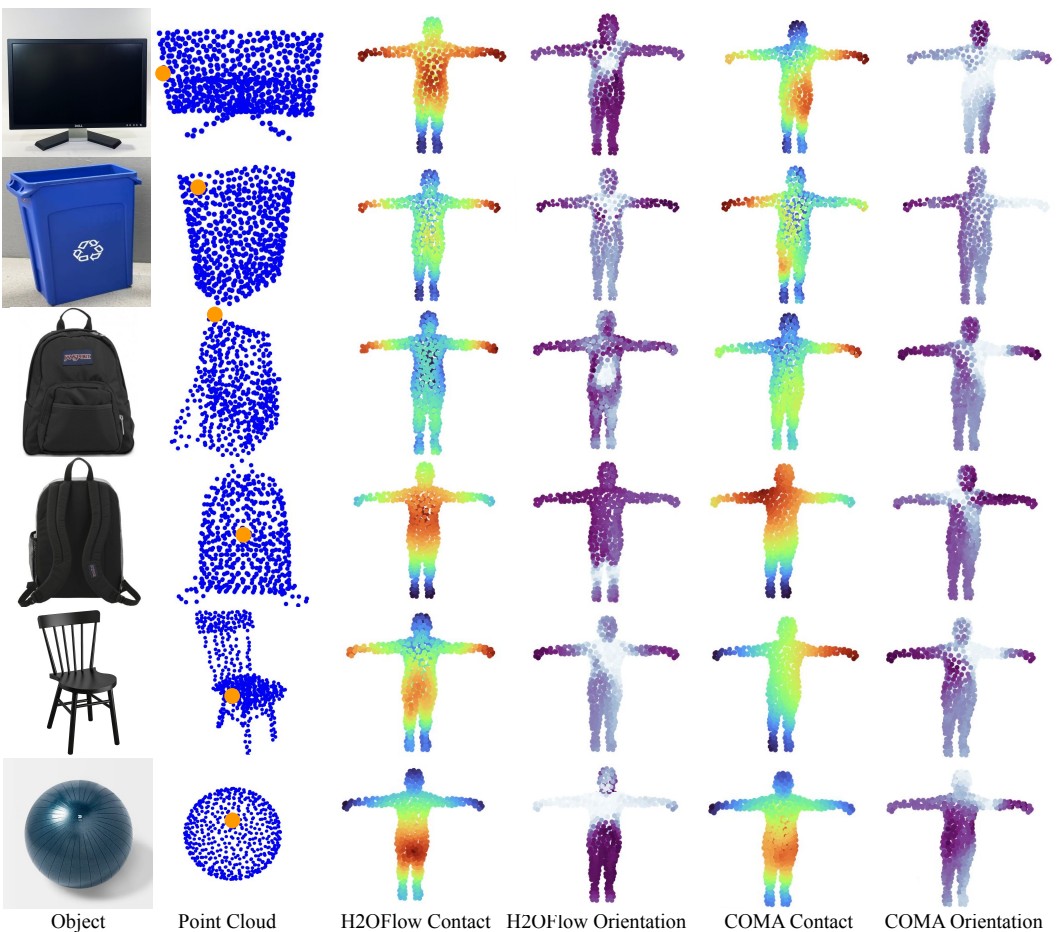

Object     Point Cloud     H2OFlow Contact    H2OFlow Orientation     COMA Contact     COMA Orientation

Figure 7: Comparison with COMA on real point clouds.

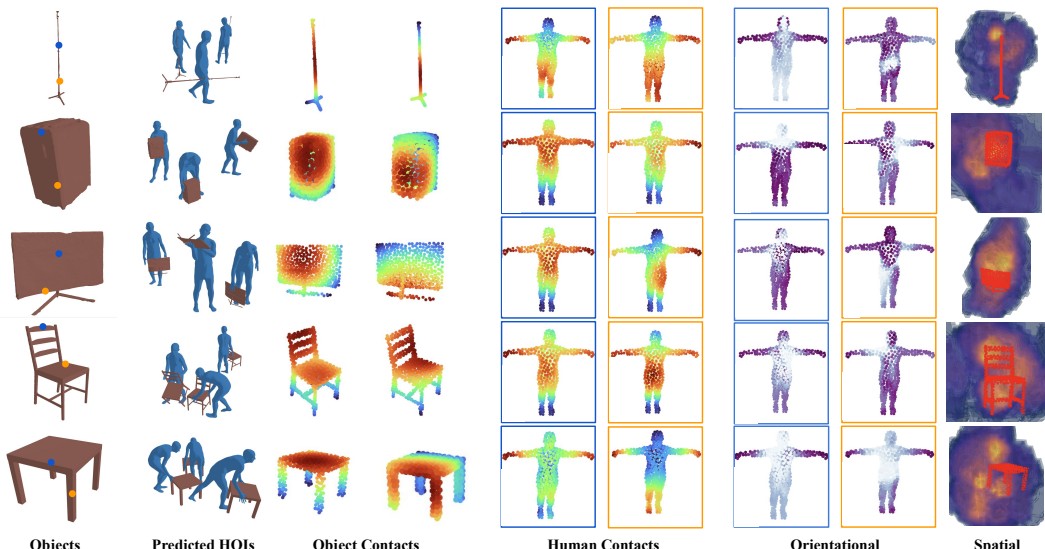

Figure 8: Visualizations of the affordances inferred from dense diffused flows prediction. H2OFlow infers diverse affordance distributions from predicted HOI samples on unseen objects.

| Baseline | SIM-H ↑ | SIM-O ↑ | MAE-H ↓ | MAE-O ↓ | Precision@K ↑ | MSE↓ |
|---|---|---|---|---|---|---|
| COMA | $52.8 \pm 1.5\%$ | $70.6 \pm 1.1\%$ | $0.13 \pm 0.04$ | $0.03 \pm 0.01$ | $72.4 \pm 5.1\%$ | $0.13 \pm 0.05$ |
| COMA-Recon | $42.5 \pm 3.2\%$ | $55.8 \pm 1.7\%$ | $0.22 \pm 0.10$ | $0.11 \pm 0.15$ | $42.4 \pm 2.2\%$ | $0.36 \pm 0.21$ |
| H2OFlow-NoFT | $55.3 \pm 3.6\%$ | $71.2 \pm 1.4\%$ | $0.13 \pm 0.03$ | $0.03 \pm 0.01$ | $72.2 \pm 1.2\%$ | $0.14 \pm 0.02$ |
| **H2OFlow-FT** | $\mathbf{74.4 \pm 1.2\%}$ | $\mathbf{80.1 \pm 1.8\%}$ | $\mathbf{0.10 \pm 0.03}$ | $\mathbf{0.02 \pm 0.01}$ | $\mathbf{79.1 \pm 5.2\%}$ | $\mathbf{0.11 \pm 0.02}$ |

Table 4: Quantitative comparisons with various baselines on BEHAVE dataset. Note that -H and -O represent human and object contact results.

are shown in Table 4. Without any fine-tuning, H2OFlow performed comparably with COMA's full-mesh version. After fine-tuning on the BEHAVE dataset, H2OFlow's performance exceeded COMA's by a noticeable margin.

## J    COMPARISONS WITH OTHER CONTACT-ONLY BASELINES

While few other methods focused on comprehensive affordances, we provide more comparisons with other contact-affordance-only baselines in Table 5. Specifically, we compare with IAGNet Yang et al. (2023) and DECO Tripathi et al. (2023) which respectively only measure contact affordances for human and object on BEHAVE test images.

## K    ABLATION ON OCCLUSION

Random masking during training lets the model accept incomplete scans. In Table 1 and Table 4, we also see that COMA struggles with outputting high-quality affordances on reconstructed meshes from partially observed point clouds, while H2OFlow is agnostic to occlusion due to training-time augmentation. New experiments also support the fact that H2OFlow is robust to out-of-distribution occlusion due to the random masking introduced in training. In Table 6, we evaluate H2OFlow's sensitivity to occlusion on test objects and show that the performance loss due to occlusion and partial observability is minimal, indicating robustness to commodity depth cameras or monocular depth-completion pipelines.

| Baseline | SIM-H ↑ | SIM-O ↑ | MAE-H ↓ | MAE-O ↓ |
|---|---|---|---|---|
| IAGNet | - | $64.34 \pm 1.2\%$ | - | $0.03 \pm 0.01$ |
| DECO | $23.18 \pm 2.1\%$ | - | $0.23 \pm 0.07$ | - |
| COMA | $52.8 \pm 1.5\%$ | $70.6 \pm 1.1\%$ | $0.13 \pm 0.04$ | $0.03 \pm 0.01$ |
| COMA-Recon | $42.5 \pm 3.2\%$ | $55.8 \pm 1.7\%$ | $0.22 \pm 0.10$ | $0.11 \pm 0.15$ |
| H2OFlow-NoFT | $55.3 \pm 3.6\%$ | $71.2 \pm 1.4\%$ | $0.13 \pm 0.03$ | $0.03 \pm 0.01$ |
| **H2OFlow-FT** | $\mathbf{74.4 \pm 1.2\%}$ | $\mathbf{80.1 \pm 1.8\%}$ | $\mathbf{0.10 \pm 0.03}$ | $\mathbf{0.02 \pm 0.01}$ |

Table 5: Quantitative comparisons of contact affordances only with various baselines on BEHAVE dataset. Note that -H and -O represent human and object contact results.

| Baseline | SIM-H↑ | SIM-O↑ | MAE-H↓ | MAE-O↓ | Precision@K↑ | MAE↓ |
|---|---|---|---|---|---|---|
| No Occl. | 72.3% | 81.0% | 0.11 | 0.07 | 75.6% | 0.12 |
| 10% Occl. | 72.3% | 80.8% | 0.12 | 0.07 | 75.5% | 0.12 |
| 30% Occl. | 71.2% | 80.7% | 0.12 | 0.08 | 75.3% | 0.13 |
| 50% Occl. | 70.9% | 79.5% | 0.13 | 0.08 | 74.2% | 0.13 |

Table 6: Performance under different occlusion levels.

## L  FLOW PREDICTION DESIGN CHOICE

An interesting question is why we learn to predict dense flows instead of SMPL parameters? We answer the question below.

**Local-geometry awareness.** A flow vector originates at every human vertex and therefore directly observes local object geometry; SMPL pose parameters do not. This makes flows more suitable for fine-grained, multimodal affordances.

**Lower computational cost.** Flow prediction needs only the object cloud and a canonical human point cloud; SMPL-parameter regression would additionally require reconstructing the human and sampling vertices — a separate task. Moreover, for orientational affordance, normal directions would have been required if we directly learned SMPL without the intermediate dense flow representation. As pointed out in COMA, this is the bottleneck for computation.

**Multi-modality.** Flows allow the diffusion model to sample multiple valid endpoints (left-/right-hand grasp, frontal/back sitting). We conducted a smaller-scale experiment, where we learn a direct SMPL predictor using diffusion conditioned on the object input. For affordance scores, direct SMPL regression does not support cross-attention weights as no per-point human information was learned during learning, so weighting is not available in aggregation here.

**Performance.** Results in Table 1 suggest that the direct SMPL formulation tends to perform a lot worse. One potential reason is that human pose parameter, especially rotation, is a lot harder to learn. Previous work Eisner et al. (2022); Pan et al. (2023); Zhang et al. (2023); Li et al. (2024b); Cai et al. (2024) on dense-flow learning designed flows to sidestep the rotation learning issue.

## M  MEMORY AND RUNTIME COMPARISONS

H2OFlow, on average, utilizes $8416 \pm 513$ MB of GPU memory and $7619 \pm 882$ MB of CPU memory. On a single V100 GPU, it takes H2OFlow $6.7 \pm 1.2$ seconds to infer affordances for an unseen point cloud. In contrast, our experiments with COMA indicate that it takes $9771 \pm 1190$ MB of GPU memory and $15812 \pm 2314$ MB of CPU memory. On a single V100 GPU, it takes COMA $65.2 \pm 2.1$ seconds to infer affordances for an unseen object. When given a point cloud, the time spent on creating a watertight mesh for COMA becomes the bottleneck. This is expected, as COMA analyzes per-pair mesh vertex normal directions and requires extensive inpainting operations. H2OFlow bypasses the large memory consumption by predicting point clouds directly.

## N  APPLICATIONS TO OTHER DOMAINS VIA DENSE OPTIMIZATION

We discuss two potential use cases of the learned diffused flows and affordance scores.

### N.1 Reconstructing Full SMPL Parameters from Dense Diffused Flows

One straightforward application is to reconstruct the full human SMPL parameters from the point cloud generated from the learned diffused flows model. Specifically, we are able to recover the full SMPL pose and shape parameters via the following optimization problem:

$$\hat{\boldsymbol{\theta}}, \hat{\boldsymbol{\beta}}, \hat{\mathbf{R}}, \hat{\mathbf{t}} = \arg\min_{\boldsymbol{\theta}, \boldsymbol{\beta}, \mathbf{R}, \mathbf{t}} \mathcal{L}(\boldsymbol{\theta}, \boldsymbol{\beta}, \mathbf{R}, \mathbf{t})$$

$$\text{with} \quad \mathcal{L} = \underbrace{\sum_{i \in \mathcal{S}} \|\mathbf{R}\, \mathbf{v}_i(\boldsymbol{\theta}, \boldsymbol{\beta}) + \mathbf{t} - \mathbf{h}_i^*\|_2^2}_{\text{vertices-reconstruction error}} + \lambda_\theta \|\boldsymbol{\theta}\|_2^2 + \lambda_\beta \|\boldsymbol{\beta}\|_2^2, \tag{10}$$

where $\mathcal{S}$ is the set of sampled vertices from the dense diffused flow model $\mathbf{v}_i(\boldsymbol{\theta}, \boldsymbol{\beta})$ is the vertex location from using the SMPL parameters. $\|\boldsymbol{\theta}\|$ and $\|\boldsymbol{\beta}\|$ act as priors because the data term alone is under-constrained when only a sparse subset of vertices are seen. Using dense diffused flow alone, we are able to reconstruct the full SMPL mesh. This again illustrates that the flows are an implicit form of affordance.

### N.2 Cross-Embodiment Reconstruction

Another interesting aspect of the affordance scores is to reconstruct different embodiments based on the predicted human point cloud. For example, suppose we are able to obtain dense cross-embodiment correspondence between a robot point $\boldsymbol{r}_k$ to human point $\boldsymbol{h}_i$, then we are able to reconstruct the full robot configuration based on the reconstructed HOI samples and affordance scores.

Specifically, we are given a set of *pre-computed human–object scores* $\left\{c_{ij}^{\text{hum}}, R_{ij}^{\text{hum}}\right\}$ that capture the contact and orientational affordances between human surface points $\boldsymbol{h}_i$ and object points $\boldsymbol{o}_j$. Our goal is to find a robot configuration that reproduces these scores as faithfully as possible.

First, we define the following parameters:

- $\Phi \in \mathbb{R}^{p_r}$ — joint-space parameters that generate nominal robot surface points $\{\boldsymbol{r}_k(\Phi)\}_{k=1}^{N_r}$ through FK function.
- $(\mathbf{R}, \mathbf{t})$ — a global rigid transform ($\mathbf{R} \in \mathrm{SO}(3)$ and $\mathbf{t} \in \mathbb{R}^3$) that aligns the robot to the human coordinate frame; the aligned points are $\boldsymbol{r}_k'(\Phi, \mathbf{R}, \mathbf{t}) = \mathbf{R}\, \boldsymbol{r}_k(\Phi) + \mathbf{t}$.
- $\mathcal{C}_{\text{robot}}$ — the feasible set defined by the robot's joint limits, self-collision constraints, and object-penetration avoidance.

**Robot-object contact score.** For each robot point $k$ and object point $j$ we define

$$c_{kj}(\Phi, \mathbf{R}, \mathbf{t}) = \exp\bigl(-\|\boldsymbol{r}_k'(\Phi, \mathbf{R}, \mathbf{t}) - \boldsymbol{o}_j\| / \tau\bigr). \tag{11}$$

The score increases as the Euclidean distance between the aligned robot point and the object point decreases.

**Robot-object orientational score.** Let $\boldsymbol{f}_i$ be the dense diffused flow attached to human point $\boldsymbol{h}_i$. We first form a unit direction vector

$$\boldsymbol{x}_{kj}(\Phi, \mathbf{R}, \mathbf{t}) = \frac{(\boldsymbol{r}_k'(\Phi, \mathbf{R}, \mathbf{t}) - \boldsymbol{o}_j) \times \boldsymbol{f}_i}{\|(\boldsymbol{r}_k'(\Phi, \mathbf{R}, \mathbf{t}) - \boldsymbol{o}_j) \times \boldsymbol{f}_i\|}, \tag{12}$$

where the cross product couples the displacement $\boldsymbol{r}_k' - \boldsymbol{o}_j$ with the flow $\boldsymbol{f}_i$. We discretize the unit sphere $\mathbb{S}^2$ into $n_b$ cells with representative normals $\{\boldsymbol{n}_n\}_{n=1}^{n_b}$ and compute the probability that $\boldsymbol{x}_{kj}$ falls into cell $n$:

$$p_{\boldsymbol{x},kj}(n; \Phi, \mathbf{R}, \mathbf{t}) \propto \exp\Bigl(-\|\boldsymbol{x}_{kj}(\Phi, \mathbf{R}, \mathbf{t}) - \boldsymbol{n}_n\|^2 / 2\sigma^2\Bigr). \tag{13}$$

The orientation score is then the negative Shannon entropy

$$R_{kj}(\Phi, \mathbf{R}, \mathbf{t}) = -\sum_{n=1}^{n_b} p_{\boldsymbol{x},kj}(n; \Phi, \mathbf{R}, \mathbf{t}) \log p_{\boldsymbol{x},kj}(n; \Phi, \mathbf{R}, \mathbf{t}). \tag{14}$$

A low entropy indicates a strongly preferred orientation and hence a large $R_{kj}$.

**Cross-embodiment matching loss.** We force the robot scores to agree with the human scores using a weighted squared loss

$$\mathcal{L}(\Phi, \mathbf{R}, \mathbf{t}) = \sum_{k=1}^{N_r} \sum_{i=1}^{N_h} M_{ki} \sum_{j=1}^{N_o} \Big[ \big( c_{kj}(\Phi, \mathbf{R}, \mathbf{t}) - c_{ij}^{\text{hum}} \big)^2$$
$$+ \lambda \big( R_{kj}(\Phi, \mathbf{R}, \mathbf{t}) - R_{ij}^{\text{hum}} \big)^2 \Big]. \tag{15}$$

The correspondence weight $M_{ki}$ transfers each human score $(i, j)$ to its associated robot point $k$.

**Optimization problem.** Our final objective is to minimize the loss (15) subject to the kinematic constraints:

$$\min_{\substack{\Phi \in \mathcal{C}_{\text{robot}}, \\ \mathbf{R} \in SO(3), \ \mathbf{t} \in \mathbb{R}^3}} \mathcal{L}(\Phi, \mathbf{R}, \mathbf{t}) \tag{16}$$

Solving (16) yields a robot pose $(\Phi^\star, \mathbf{R}^\star, \mathbf{t}^\star)$ whose contact and orientational affordance fields best imitate those observed for the human demonstrator, while remaining physically feasible for the robot.

## O REAL-WORLD QUANTITATIVE RESULTS

We design a set of experiments to quantitatively assess H2OFlow's performance on real-world objects. While real-world ground truth is not available, we devise two complementary lines of metrics to compare baseline performance. We select six real-world objects that have similar simulated counterparts in the OMOMO and BEHAVE datasets. Each real object point cloud is aligned to a canonical object frame using ICP with its closest simulated counterpart.

We report two types of quantitative metrics: (1) **SMPL-based metrics**, which compare reconstructed goal human poses against CHOIS-generated synthetic human-object interactions, and (2) **affordance-based metrics**, which estimate relative plausibility and spatial consistency of predicted affordances. Since affordance ground truth is not available in real-world data, these are used only as complementary evidence.

Given a real object $O$, we sample $N$ goal human configurations from H2OFlow and reconstruct corresponding SMPL poses $\mathcal{S}_{\text{H2O}} = \{S_{\text{H2O}}^{(n)}\}_{n=1}^N$ using the optimization defined in Eq.10. For the COMA baseline, we use $N$ generated SMPLs inferred from multi-view 2D renderings of the same object. We collect $M$ reference SMPL poses from CHOIS on a geometrically similar simulated object, denoted $\mathcal{S}_{\text{CHOIS}} = \{S_{\text{CHOIS}}^{(m)}\}_{m=1}^M$. All SMPLs are centered in the object coordinate frame with pelvis translation removed.

### O.1 DEFINING METRICS FOR REAL-WORLD EXPERIMENTS

We compute three set-to-set distances between generated and reference SMPL sets. First, Minimum Matching Distance (MMD) measures the average minimum per-sample joint distances $d(S, S') = \frac{1}{J} \sum_{j=1}^J \|J_j(S) - J_j(S')\|_2$.

$$\text{MMD}(\mathcal{S}, \mathcal{S}_{\text{ref}}) = \frac{1}{|\mathcal{S}|} \sum_{S \in \mathcal{S}} \min_{S' \in \mathcal{S}_{\text{ref}}} d(S, S'), \tag{17}$$

Next, Coverage measures the percentage of candidate set that covers the modes of reference set:

$$\text{COV}_\varepsilon = \frac{|\{S' \in \mathcal{S}_{\text{ref}} : \exists S \in \mathcal{S}, \ d(S, S') \leq \varepsilon\}|}{|\mathcal{S}_{\text{ref}}|}, \tag{18}$$

Lastly, Frechet Pose Distance (FPD) measures the distributional similarity between the prediction sets:

$$\text{FPD} = \|\mu_c - \mu_r\|_2^2 + \text{Tr}(\Sigma_c + \Sigma_r - 2(\Sigma_c^{1/2} \Sigma_r \Sigma_c^{1/2})^{1/2}), \tag{19}$$

where $J_j(S)$ denotes the 3D joint positions of pose $S$, and $(\mu_c, \Sigma_c)$, $(\mu_r, \Sigma_r)$ are the means and covariances of joint vectors from the candidate and reference sets, respectively. Lower MMD and

| Method | MMD ↓ | COV@15cm (%) ↑ | FPD ↓ | SIM-H (%) ↑ | MAE-H↓ | Precision@K (%)↑ | MSE ↓ |
|---|---|---|---|---|---|---|---|
| COMA | 12.8 | 43.2 | 61.5 | 49.7 | 0.23 | 40.9 | 0.21 |
| **H2OFlow** | **8.9** | **64.5** | **47.8** | **68.6** | **0.13** | **71.3** | **0.13** |
| **H2OFlow (None)** | 10.6 | 58.9 | 50.6 | 65.4 | 0.16 | 67.2 | 0.16 |
| **H2OFlow (Light)** | 9.2 | 63.1 | 48.6 | 68.0 | 0.14 | 70.8 | 0.14 |
| **H2OFlow (Medium)** | **8.9** | **64.5** | **47.8** | **68.6** | **0.13** | **71.3** | **0.13** |
| **H2OFlow (Aggressive)** | 9.9 | 60.2 | 49.5 | 65.9 | 0.15 | 68.1 | 0.17 |

Table 7: Quantitative comparison of SMPL-based distances between real-world predictions and CHOIS reference poses. Lower MMD/FPD and higher Coverage indicate better alignment with synthetic interaction distributions. "Light/Medium/Aggressive" correspond to the presets in Section O.2.

FPD and higher $COV_\varepsilon$ indicate closer alignment between predicted and synthetic interaction distributions.

We additionally report similarity between predicted contact and spatial occupancy distributions derived from real scans and the corresponding simulated object using the same metrics used in simulated experiments. These metrics capture how well the learned affordance distributions transfer to real geometry.

Across six real-world objects, H2OFlow achieves significantly lower MMD/FPD and higher coverage than COMA, confirming its ability to generalize to real-world objects without manual labeling.

## O.2 ROBUSTNESS TO NOISE LEVEL

Our real-world captures undergo post-processing, which can yield relatively clean point clouds. To quantify robustness, we test H2OFlow on *noisy* single-object point clouds and define a reproducible denoising pipeline with controllable strength. We assume single, segmented object point clouds, captured via a commodity RGB-D camera (in our case, an iPhone camera).

We adopt a standard point-cloud pipeline available in Open3D Library. Each stage and its control parameter is listed so denoising strength is explicit and reproducible.

1. **Statistical Outlier Removal (SOR):** `statistical_outlier_removal(nb_neighbors = ` $k$ `, std_ratio = ` $r$ `)`, which removes isolated points whose mean neighbor distance exceeds $r$ standard deviations. *Controls:* $k$ (neighbors), $r$ (aggressiveness).

2. **Radius Outlier Removal (ROR):** `radius_outlier_removal(nb_points = ` $m$ `, radius = ` $R$ `)`, which prunes sparse regions lacking $m$ neighbors within radius $R$. *Controls:* $m$, $R$ (m).

We then apply FPS to the cleaned point clouds, yielding a denoised, subsampled result. We define four presets that sweep aggressiveness:

| Preset | $k$ | $r$ | $m$ | $R$ | Visualization |
|--------|-----|-----|-----|-----|---------------|
| **None** | – | – | – | – |  |
| **Light** | 20 | 2.0 | 8 | 0.01 |  |
| **Medium** | 30 | 1.5 | 12 | 0.012 |  |
| **Aggressive** | 50 | 1.0 | 16 | 0.015 |  |

In our evaluation, we use the same H2OFlow weights across all conditions. For each object, we use RealityKit to collect raw noisy scan, denoised with **None/Light/Medium/Aggressive**. We use the same real-world objects in Section O.1 with the same real-world evaluation protocols.

H2OFlow, by default, uses the medium denoising setting. As shown in Table 7, it is point-cloud–native and remains stable under Light and Medium denoising. By contrast, the Aggressive preset can over-prune thin structures (*e.g.*, handles, rims), degrading affordance performance near those parts and increasing collisions during pose selection. We also observe that the drop of performance is not substantial: although both None and Aggressive settings reduce performance relative to Light/Medium, they still outperform the COMA baseline, underscoring the effectiveness of flows as the intermediate representation.

# P    SCALABILITY

## P.1    SCALING VIA DATA AUGMENTATION

For objects like chairs, the dominant affordance is often *hips-on-seat* (sitting) rather than hand-mediated manipulation. Because our training set is synthesized from CHOIS (object motion–guided interactions), it over-represents *moving/lifting* patterns. Fortunately, H2OFlow's training recipe is model-agnostic and point-cloud–centric, so we can *expand* the HOI source models to better cover everyday *usage* (sit, lean, rest, place, open/close) and retrain the same dense-flow learner.

We augment the synthetic HOI pool with recent 3D HOI generation models that explicitly produce usage-centric human–object interactions such as InteractAnything Zhang et al. (2025), HOI-PAGE Li & Dai (2025), and PICO Cseke et al. (2025). All generators are standardized by our existing pre-processing: mesh $\rightarrow$ FPS subsampling $\rightarrow$ paired point clouds, identical to our current pipeline. No watertight meshes or normals are required downstream. By adding more training data, we effectively expand the actions used: *sit on*, *perch*, *lean back*, *rest arm*, *place on seat*, *kneel*, *step onto*, *lie on*... We assemble a balanced mixture of *manipulation* and *usage* episodes per category and filter collisions or self-penetrations. We keep the same object splits (train/test categories) as in the main paper. We retrain the same DiT flow model with identical losses/hyperparameters; only the HOI source distribution changes. Dense diffused flows remain the intermediate representation, preserving compatibility with our affordance inference.

As the chair example (same real-world chair from Figure 5) shows in Figure 9, without training data augmentation from Zhang et al. (2025), the contact affordance mostly concentrates on the hands areas with limited contact in the hip area due to the fact that most chair interactions from CHOIS

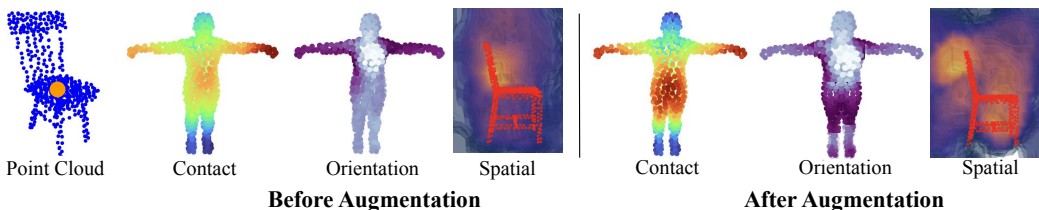

Figure 9: Comparison of the three affordance representations before and after dataset augmentation with *usage* data. After augmentation, we observe more symmetry and meaningful interaction patterns that reflect actual object usage (*e.g.*, hip-on-seat).

data are *moving* the chair. Similarly, for orientation, not much trend is seen in the hip area and for spatial affordance, the occupancy is concentrated at the back of the chair, indicating how a human moves the chair. While the above predictions are valid for *moving* the chair, more expressive affordances that indicate *sitting on* the chair are more informative. Augmented with synthetic data generated from Zhang et al. (2025), we have more HOI meshes representing sitting on the chair in the training dataset. Thus, the flow predictor will learn to predict both *moving* and *using* the chair. After this augmentation, we see a bi-modal concentration of contact affordance in both hands and hip areas, indicating the model now learns to output contact based on actual usage. For the orientation affordance, we now see better symmetry as well as concentrated orientational pattern in the legs area. Lastly, for the spatial affordance, the front side of the chair is now more frequently occupied.

Because H2OFlow learns from point-cloud flows rather than annotation-heavy labels or normals, broadening generators directly broadens learned *usage* affordances (*e.g.*, *hips-on-seat*) without architectural change.

## P.2    PROMPT-CONDITIONED DENSE DIFFUSED FLOWS

Beyond dataset diversification, we can *condition* the dense-flow generator on a textual intent ("sit on the chair"). This steers sampling toward usage-consistent interactions (seat occupancy, torso orientation) at test time, even for unseen objects, while keeping H2OFlow's point-based formulation intact.

Let $t$ be a tokenized prompt. We encode text with a frozen CLIP text encoder Radford et al. (2021) $E_{\text{text}}(t) \in \mathbb{R}^{L \times d}$. We then condition the DiT backbone used for dense-flow denoising with cross-attention adapters: in each DiT block, after self-attention on the joint human-flow tokens (as in the main model), add a cross-attention from human-flow tokens to text tokens. Object tokens remain as in the main model (object→human cross-attention). We also retain the same noise-prediction targets and hybrid diffusion loss as our current DiT.

As standard in conditional diffusion Ho & Salimans (2022), we drop text with probability $p_{\text{drop}}$ during training and learn unconditional/conditional branches jointly. At inference, we sample flows with guidance scale $\gamma$: $\hat{\epsilon}_\theta = (1 + \gamma)\,\epsilon_\theta(F_t \,|\, O, t, \varnothing) - \gamma\,\epsilon_\theta(F_t \,|\, O, t, E_{\text{text}}(t))$.

During training, we attach short textual prompts to each synthetic HOI episode and automatically normalize existing action descriptions into concise templates (e.g., "*sit on chair*", "*lean back on backrest*"), preserving the same category splits. At test time, language steers the sampled human goal configurations $\boldsymbol{H} = \boldsymbol{H}_0 + \boldsymbol{F}$; our affordance inference (contact $c_{ij}$, orientational $R_{ij}$, spatial $S_{ij}$) remains unchanged.

In terms of implementation, the change is minimal, as only a few pieces in the DiT are changed.

```
class DiTBlock(nn.Module):
    def __init__(self, hidden_size, num_heads:
        super().__init__()
        self.self_attn    = SA(hidden_size, num_heads)
        self.cross_attn_o = CA(hidden_size, num_heads)
        self.mlp = Mlp(hidden_size, int(hidden_size * 4))

        # adaLN-Zero modulation
        # outputs shifts/scales/gates for 3 submodules = 9 chunks
```

```python
        self.adaLN = nn.Sequential(
            nn.SiLU(),
            nn.Linear(hidden_size, 9 * hidden_size, bias=True)
        )

    def forward(self, x_hf, y_obj, cond, x_pos=None, y_pos=None):
        (sh_msa, sc_msa, gt_msa,
         sh_mo,  sc_mo,   gt_mo,
         sh_mlp, sc_mlp, gt_mlp) = self.adaLN(cond).chunk(9, dim=1)

        x = modulate(self.norm_msa(x_hf), sh_msa, sc_msa)
        x = x + gt_msa.unsqueeze(1) * self.self_attn(
            query=x, key=x, value=x, rotary_pe=(x_pos, x_pos)
        )[0]

        x_o = modulate(self.norm_mca_o(x), sh_mo, sc_mo)
        x = x + gt_mo.unsqueeze(1) * self.cross_attn_o(
            query=x_o, key=y_obj, value=y_obj, rotary_pe=(x_pos, y_pos)
        )[0]

        x_ff = modulate(self.norm_mlp(x), sh_mlp, sc_mlp)
        x = x + gt_mlp.unsqueeze(1) * self.mlp(x_ff)

        return x
```

Figure 10: Original H2OFlow's DiT Block Implementation in PyTorch

```python
class DiTBlockText(nn.Module):
    def __init__(self, hidden_size, num_heads):
        super().__init__()
        self.self_attn     = SA(hidden_size, num_heads)
        self.cross_attn_o = CA(hidden_size, num_heads)
        self.cross_attn_t = CA(hidden_size, num_heads)
        self.mlp = Mlp(hidden_size, int(hidden_size * 4))

        # adaLN-Zero modulation
        # outputs shifts/scales/gates for 4 submodules = 12 chunks
        self.adaLN = nn.Sequential(
            nn.SiLU(),
            nn.Linear(hidden_size, 12 * hidden_size, bias=True)
        )

    def forward(self, x_hf, y_obj, z_txt, cond, x_pos, y_pos, z_pos):
        (sh_msa, sc_msa, gt_msa,
         sh_mo,  sc_mo,   gt_mo,
         sh_mt,  sc_mt,   gt_mt,
         sh_mlp, sc_mlp, gt_mlp) = self.adaLN(cond).chunk(12, dim=1)
        x = modulate(self.norm_msa(x_hf), sh_msa, sc_msa)
        x = x + gt_msa.unsqueeze(1) * self.self_attn(
            query=x, key=x, value=x, rotary_pe=(x_pos, x_pos)
        )[0]

        x_o = modulate(self.norm_mca_o(x), sh_mo, sc_mo)
        x = x + gt_mo.unsqueeze(1) * self.cross_attn_o(
            query=x_o, key=y_obj, value=y_obj, rotary_pe=(x_pos, y_pos)
        )[0]

        x_t = modulate(self.norm_mca_t(x), sh_mt, sc_mt)
        x = x + gt_mt.unsqueeze(1) * self.cross_attn_t(
            query=x_t, key=z_txt, value=z_txt, rotary_pe=(x_pos, z_pos)
        )[0]

        x_ff = modulate(self.norm_mlp(x), sh_mlp, sc_mlp)
        x = x + gt_mlp.unsqueeze(1) * self.mlp(x_ff)
        return x
```

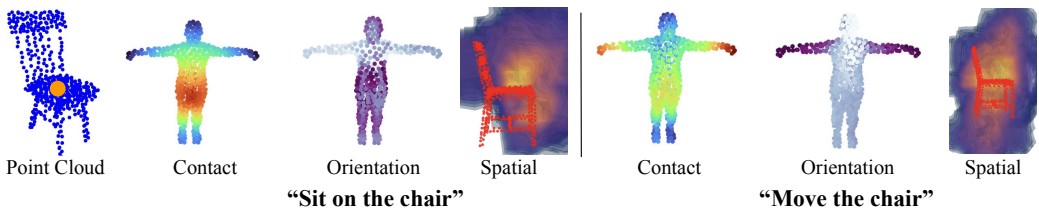

Figure 12: Chair example of prompt-conditioned H2OFlow

Figure 11: Prompt-Conditioned H2OFlow's DiT Block Implementation in PyTorch

As shown in the implementation of Figure 10 and Figure 11, with minimal changes to the DiT block architecture, we are able to condition the output flows on the tokenized prompts. In Figure 12, we use the same example as above but test it on the text-conditioned H2OFlow. As one can observe, with the prompt conditioning, the two different usages of the chair yield really different affordances. The contact now is not bimodal between hands and hips and is now concentrated on one body part based on the usage. Similarly, for the orientation affordance, we observe high concentration on hips when sitting and on arms when moving. More interestingly, we see a sitting silhouette for sitting and a standing silhouette for moving.

Thus, a small, modular text head yields *promptable* dense flows that align with language-specified affordances, while preserving the core point-cloud training and inference of H2OFlow.

## Q  LIMITATIONS

While H2OFlow learns comprehensive affordances from synthetic data and generalizes to unseen objects' noisy point clouds, it has a few limitations. First, the underlying generative model does not have a sufficiently large variety of objects–more fine-grained interactions on smaller, articulated objects are not captured. While H2OFlow can learn from arbitrary HOI samples, there are limited foundation models and datasets on such fine-grained HOI tasks. Second, H2OFlow could be extended to physical robots, by warmstarting the manipulation policies based on the affordance score while constructing a correspondence map between human point clouds and robot points. We leave this extension to future work.

## R  LLM USAGE

We primarily used LLMs to check grammar and spelling. We also used LLMs for formatting tables and figures.

