# OpenReview forum: "H2OFlow: Grounding Human-Object Affordances with 3D Generative Models and Dense Diffused Flows"
_ICLR.cc/2026/Conference — ICLR 2026 Poster_

### Official Review · Reviewer_TEPZ · 2025-10-23

**Soundness:** 3
**Presentation:** 3
**Contribution:** 3
**Rating:** 6
**Confidence:** 3

**Summary:**

The paper proposes a pipeline for learning object affordances from 3D Human-Object interactions obtained using 3D generative AI. First, the method uses a pre-trained generation pipeline to generate human-object interaction motions, then uses it to train a diffusion model that, given an object, predicts plausible humans interacting with it using Dense Flow. The paper studies three kinds of affordances: contact, spatial, and orientational-based, and compare against several baselines and objects with geometry unseen at training time, showing good generalization.

**Strengths:**

- I found the paper interesting, proposing a direction orthogonal to the ones in the previous methods. The study of different kinds of affordances could serve to define object classes more effectively based on their spatial and orientation qualities.
- Improvement over competitors is significant, and the results on real objects acquired with a phone camera look exciting

**Weaknesses:**

- Generate interaction data using a pre-trained 3D network specifically designed for generating 3D HOI might actually incorporate biases from such methods. For example, as mentioned in the paper, the majority of methods focus on contact-based interactions, and hence, the generated data are also affected. Additionally, one might wonder why not use the original datasets used to train CHOIS as training data for the Dense Flow. Finally, the Dense Flow module works at a frame level, while CHOIS generates sequences, which can be quite redundant in terms of nearby frames.

- The point clouds used in the real-world setting look particularly clean. I assume it is part of the post-processing done by the capturing device, but it would also be useful to test on more noisy data, such as point clouds from kinect fusions, as those available in the BEHAVE dataset.

- Qualitative results are not well presented. Figure 3 illustrates, for example, contact over the object; however, although the table is symmetric, the legs have different affordances, highlighting a skewed distribution in the dataset. On the human body, contact is spread on legs, which I do not find intuitive. Orientational and spatial are difficult to decode. I would suggest reporting a colormap for the visualized colors, describing better what is visualized, and using a mesh when available (e.g., SMPL)

**Questions:**

- Could you comment on your choice of using generative AI, instead of directly using the data on which they are trained?
- Is it possible to provide an example on a more noisy point cloud, maybe discussing how much outliers cause degradation in the network prediction?

---

> ### Author Response · Authors · 2025-11-19
> **(1/3) Thank you, please see below for response. We have added new ablations and more results in the revised manuscript (**see highlighted text, new experiments in Section 5,  new Appendix P, and new Appendix Q: Scalability**)**
>
> We sincerely thank Reviewer TEPZ for the constructive and detailed review. We appreciate the recognition of the **impact of dense-diffused-flow-based comprehensive affordance** on object modeling and the empirical **robustness** of H2OFlow. The comment on the strong **novelty** of the work resonates with other reviewers’ comments as well. We now address the reviewer's questions.
>
> > Why not train directly on the generator’s original datasets?
>
> State-of-the-art HOI synthesis works such as CHOIS are trained on object meshes (optionally with sparse waypoints) and then **synthesize human poses on top of them**. The upstream training data (objects) typically (i) emphasize contact endpoints or sparse kinematic cues, (ii) lack dense, temporally consistent supervision for orientation and spatial occupancy, and (iii) often do not provide paired meshes suitable for point-cloud learning. In short, those original datasets do not contain the comprehensive HOI signals that our method learns to infer.
>
> > On “contact-focused” upstream generators.
>
> We agree that many HOI generators emphasize contact; however, in affordance learning, “contact” is a *necessary but not sufficient* signal. Even contact-rich interactions depend critically on (i) orientation (e.g., wrist/forearm alignment to a handle plane) and (ii) spatial occupancy (e.g., approaching from the front/side to avoid collisions). In our new experiments, provided in **Section 5: Affordance Types Ablation** of the main text and **Appendix P: Are Non-Contact Affordances Useful?**, our ablations isolate these effects, and we measure the performance of using Contact-only $\mathbf{C}$, Contact+Orientational $\mathbf{C{+}O}$, Contact+Spatial $\mathbf{C{+}S}$, Full $\mathbf{C{+}O{+}S}$, plus **shuffle controls**, where we keep C but shuffle O and S on downstream tasks.
>   - Results suggest that against $\mathbf{C}$, $\mathbf{C{+}O}$ and $\mathbf{C{+}S}$ significantly improve the metrics,  and $\mathbf{C{+}O{+}S}$ yields further gains.
>   - Shuffled controls eliminate these improvements, confirming that structured orientational and spatial affordances indeed improve performance for affordance learning, not merely because of additional feature capacity.
>
> In other words, **beyond-contact affordances enhance contact-centric tasks** by disambiguating how and from where contact should occur.
>
> > Why a generative model as the upstream source? Why multiple frames?
>
>  Using a pretrained HOI generator gives us a **large, controllable supply** of 3D HOI sequences: each sequence yields many frames that we convert to paired point clouds, perfectly aligned with our point-based, mesh-agnostic training. We can sample unbounded interactions per object category and systematically sweep object poses, sizes, and scene layouts.
>
> Our current scheme is that we sample 100 sequences per object, and we then choose the middle 200 key-frames per sequence and each frame is a datapoint. This ensures that our training data is **as diverse as possible** from a single data source. This also ensures that our training dataset is **easily extendable.**
>   - As shown in the new **Appendix Q**, we augment the training dataset with more latest 3D HOI synthesis models. This **plug-and-play** nature also **eliminates the reliance** on any single data source and enables compositional diversity, as we can prompt generators for usage-centric behaviors (e.g., sit/lean/rest/place) that are under-represented or absent in the original training data.
>
> Another reason is the direct compatibility with our representation: our target is a dense, probabilistic flow field over point clouds; synthesized HOIs provide temporally coherent 3D supervision that maps cleanly to our diffusion objective, without reconstructing surfaces or normals. Lastly, the original (upstream) data is limited in size/scope and is costly to expand; sampling from trained generators is effectively free, letting us iterate on coverage quickly.

---

> ### Author Response · Authors · 2025-11-19
> **(2/3) Please see Appendix O.2 and Appendix Q for new results and figures for this section**
>
> > On noisy point clouds
>
> During training, we apply random rotation to the objects and random occlusion as augmentation in order to ensure robustness to real-world variability (**Appendix E**), which ensures the robustness to occlusion in the real world, as opposed to COMA that requires a clean mesh of the object. In **Table 6 of Appendix K**, which was provided in the original submission, we evaluate H2OFlow's sensitivity to occlusion on OMOMO test objects and show that the performance loss due to occlusion and partial observability is minimal, indicating robustness to commodity depth cameras or monocular depth-completion pipelines.
>
> In the newly provided **Appendix O.2**, we also conduct new experiments on the robustness of varied levels of denoising of real captured point clouds. The detailed description of the experiment is in the revised manuscript, but we summarize some key points here.
>
> - To clean the point clouds, we adopt a standard point-cloud pipeline available in Open3D Library. We control the strength of each stage and its control parameter to make the process adjustable and reproducible, where we define four levels of denoising aggressiveness (we also provide some **visualizations** of the cleaned point clouds in **Appendix O.2**).
> - H2OFlow, by default, uses the medium denoising setting.
> - As shown in the new **Table 7**, it remains stable under Light and Medium denoising. We observe that the drop of performance is **not substantial**: although both None and Aggressive settings reduce performance relative to Light/Medium, they still **outperform the COMA baseline**, underscoring the effectiveness of flows as the intermediate representation.
>
> Thus, H2OFlow’s affordance quality and downstream selection remain stable under noise levels; performance degrades gracefully under limited and extreme pruning and still exceeds COMA across metrics. This supports our claim that learning dense flows over point clouds is an effective inductive bias for noisy, partial real-world inputs.
>
> > On the presentation of results
>
> We appreciate the reviewer for pointing out the presentation issue. In the new **Figure 4* of the revised manuscript, we have added **colormaps** for your reference.
>
> Fundamentally, our primary representation for contact is the per-vertex maps on the body, as the designed affordances are human-centric: *conditioned on the object point, what are the plausible distributions for human points?* The “object affordance” panel is a *derived* visualization obtained by **transposing the human-object contact affordance matrix** to show where the human-side evidence concentrates on the object. It is *diagnostic*, not an additional supervised target, and does not affect training or the human-side affordances.
>
> - In fact, the human-side symmetry is improved by the usage of cross-attention weight (see **Figure 5(a) and Cross-Attention Weights Ablation in Section 5**).
>
> Regarding the specific asymmetry in this case, since we canonicalize each object’s pose, per-leg differences reflect the empirical interaction distribution in the synthetic HOI sequences. This is indeed a dataset skew signal surfacing in the visualization. As shown in the new **Appendix Q** of the revised manuscript, H2OFlow does **scale well with the diversity** of the synthesized HOI meshes, provided by more recent 3D generative models. With this scalability, H2OFlow should be able to learn such symmetry from more data. Moreover, without a new data source, we could apply random left–right mirroring of object coordinates to decorrelate fixed frame biases. We could also register symmetric parts (e.g., four table legs) and report both the raw map and a symmetrized version as the group-average over symmetric correspondences. We appreciate the reviewer for pointing this out and would like to leave it to future work.
>
> For the contact heatmap over the human legs, in several generated interaction families, humans *kick, nudge, or brace** the table with a shin/foot to move it. Those frames generate leg contact that appears in the aggregated contact map. This is **particularly interesting** as it reveals a **less common, yet potentially useful**, affordance, and is **well-captured** by the flow-based model. We conjecture that such an ability to find less common affordances as modes in an affordance score distribution is *particularly useful* for downstream spatial AI tasks.

---

> ### Author Response · Authors · 2025-11-19
> **(3/3)**
>
> We addressed the questions by (i) justifying the usage of 3D generative model, (ii) providing additional experiments on denoising levels, (iii) justifying the human-centric design and scalability of H2OFlow to account for potential symmetry in the model, and (iv) justifying H2OFlow’s ability to discover less common affordances. These changes strengthen the paper while preserving its core contribution: **learning comprehensive, robust 3D affordances from point clouds via dense diffused flows, scalable to new generators, tasks, and semantic controls**. We hope these revisions clarify our contributions and demonstrate the robustness, generalizability, and scalability of H2OFlow. Please kindly consider raising the score if your concerns and questions are addressed.

---

> > ### Comment · Reviewer_TEPZ · 2025-11-23
> > **rebuttal response**
> >
> > I thank the authors for their comprehensive reply. I have carefully reviewed their answers and the other reviews. I have just a couple of follow-up comments:
> >
> > 1) Training on upstream data: I see the author's point, but I am slightly confused. The first two points ("emphasize contact endpoints or sparse kinematic cues" and "lack of dense supervision") suggest that such problems are propagated to the generative model used to obtain the training data. If the learning mitigates such an issue, I do not see why the method proposed in the paper should not be able to achieve the same result. For the third point ("often do not provide paired meshes suitable for point-cloud learning"), it does not really seem to be the case for CHOIS, which is trained on the FullBodyManipulation dataset, which does have paired meshes. Or am I missing something in the argument?
> >
> > 2) Real point cloud: I appreciate the further result. From the text, I get that the luggage is scanned using an iPhone. The pointcloud still looks particularly clean (e.g., no outliers, no clutter like ground). I believe the provided example could be ok, but I suggest outlining more details about the capturing procedure.
> >
> > 3) Frame selection: Using the central 200 continuous key frames of a sequence looks redundant, as probably nearby frames are almost identical. I do not consider this point particularly critical, but I would suggest considering validating the need for all of them against downsampling (e.g., 1/2, 1/5, 1/10).
> >
> > 4) Symmetrical skewing: Although this might seem a minor problem, I wonder how this affects the overall evaluation. If there is a symmetrical assumption, one could also mirror the output affordance accordingly to the symmetrical planes, and obtain a representation that actually fixes any skewness in the data. Notice, however, that such an assumption is not always verified: cars are symmetric outside, but mounting on the left or right has a different affordance meaning. Similar for the use of some controllers and joysticks, where buttons are not symmetrical (and hence, symmetrical flips at training time generate data that do not reflect the real-world distribution). Luggage and boxes have an upward orientation that is not evident from the outside. Although it is a problem commonly overlooked in the literature, observing the non-symmetrical affordance warrants a discussion in the paper. I also believe that affordance completion by symmetrical flip at inference time (if symmetric assumption is retained) would be feasible within the scope of the paper.
> >
> > I do not have further questions. I reassure the author that, after considering the rebuttal and discussion with other reviewers, I will update my score accordingly.

---

> > > ### Author Response · Authors · 2025-11-24
> > > **Super helpful follow-up — thank you.**
> > >
> > > > Training on upstream data.
> > >
> > > We appreciate this question raised by the reviewer. We think there was a slight confusion over the term "upstream data." In our response, we were under the impression that it referred to the **objects meshes** that upstream methods, which are not the synthetic sequences the generator produces that we use to train Dense Flow. So we rely on generated sequences, not the generator’s original training sets (object meshes), because they are scalable, point-cloud–friendly, and contain the temporal variation needed to model distributions beyond contact. About the potentially propagated problems: even if a generator was trained with contact-heavy supervision, its outputs contain continuous motion and varied relative placements. Our dense diffused flows learn a distribution over these local geometric relations; orientation/occupancy come from statistics over sampled flows, not from explicit upstream labels.
> > >
> > > > Real point cloud
> > >
> > > We appreciate the reviewer for acknowledging the latest results. There's a built-in segmentation step run on the website. We will document this in the final version of the paper.
> > >
> > > > Frame selection
> > >
> > > We appreciate the reviewer for pointing out a potential ablation test. While we choose the center 2k frames to ensure diversity, we do acknowledge the potential improvement by reducing the number of frames via downsampling. We are actively running the experiments right now and will update the Appendix in the final version of the paper.
> > >
> > > > Symmetrical skewing
> > >
> > > Our method does not assume objects are symmetric; human-side symmetric affordances are the optimized target via attention weights, and object-side visualizations are diagnostic. However, we do acknowledge that some categories have strong geometric symmetries (e.g., 4-leg tables), while others are semantically asymmetric despite geometric symmetry (e.g., game controllers with non-symmetric button maps).
> > >
> > > We have added a new experiment to compare **object-side contact affordance** with and without symmetry completion. For the symmetric objects category, such as tables, we conduct symmetry-aware evaluation by estimating a candidate mirror plane and report both Raw and Symmetrized results. To estimate the mirror plan, we do a RANSAC-based estimation:
> > >
> > > - RANSAC plane fit on object point cloud; keep near-vertical planes.
> > > - Pick the plane that maximizes ICP self-alignment objective value defined by the Chamfer distance reduction after mirroring and ICP.
> > > - Once we obtain the plane, we mirror the points and register the nearest-neighbor with the same affordance value.
> > > - This results in a two-way mirroring affordance, and we report *the best-of-two* when reporting the symmetrized results.
> > >
> > > On our simulated test objects, we select the symmetric categories and report the SIM and MAE results.
> > >
> > > | Categories |Dining Tables | Small Tables | Chairs | Box |
> > > |--------------|----------------|----------------|-------|-------|
> > > |Raw SIM $\uparrow$ (%)|       79.0$\pm$3.2                |            82.2$\pm$2.4         |       80.9$\pm$3.4      |   79.3$\pm$1.8       |
> > > |Symmetrized SIM $\uparrow$ (%)|      80.2$\pm$1.2         |      84.3$\pm$1.8      |      81.1$\pm$2.1      |        79.9$\pm$1.2  |
> > > | Raw MAE $\downarrow$|   0.11$\pm$0.04    |     0.06$\pm$0.01    |   0.07$\pm$0.02        |   0.09$\pm$0.03   |
> > > |Symmetrized MAE $\downarrow$ |  0.09$\pm$0.02     |   0.06$\pm$0.01    |     0.07$\pm$0.01      |    0.09$\pm$0.02        |
> > >
> > > Results indicate that after applying symmetry mapping, we consistently achieve slightly better results and lower variances. This indicates a potential improvement by exploiting the object symmetry to achieve better affordance learning results. This direction, as well as the symmetry-aware evaluation, is a great next step for the next version of H2OFlow, and we sincerely appreciate the reviewer for pointing this out.

---

### Official Review · Reviewer_Y1sC · 2025-10-25

**Soundness:** 3
**Presentation:** 3
**Contribution:** 3
**Rating:** 6
**Confidence:** 4

**Summary:**

This paper proposes H2OFlow, a novel framework for learning comprehensive 3D human-object interaction (HOI) affordances, encompassing contact, orientation, and spatial occupancy. The method leverages a pre-trained 3D generative model to create a synthetic dataset, thus avoiding manual annotation. Its core innovation is a diffusion model trained to predict dense flow, a probabilistic, point-based representation of human interaction poses conditioned on an object's point cloud. In inference, sampling these flows generates a distribution of plausible interactions, from which the three affordance maps are statistically derived.

**Strengths:**

* The paper introduces an innovative framework that learns comprehensive affordances from synthetic HOI samples generated by 3D generative models. This approach cleverly eliminates the need for manual annotation and avoids the error-prone 2D-to-3D uplifting process used in prior work.
* A key contribution is the use of "dense diffused flows" as a probabilistic, point-based representation for human interaction. This design, learned via a diffusion model, elegantly circumvents the dependency on manifold meshes and is directly responsible for the method's outstanding robustness to the noisy and partial point clouds typical of real-world sensors. The superiority of this representation is further validated by strong experimental results and thorough ablations.

**Weaknesses:**

* The framework's performance is fundamentally tethered to the quality and diversity of a single upstream generative model (CHOIS). The paper appears to neglect any strategy for analyzing, filtering, or augmenting synthetic data. This raises critical questions: How does the model mitigate the risk of inheriting and amplifying potential biases from its sole data source—such as limitations in interaction patterns, insufficient object diversity, and unnatural poses? What are the upper limits of the affordance knowledge defined by CHOIS?
* The proposed method excels at exploring the general distribution of possible geometric interactions but lacks semantic control or decoupling mechanisms. This leads to two issues: (a) whether it is conditioned on specific task instructions (real-world scenarios should prioritize concrete interactions); (b) whether it can distinguish between objects that are geometrically similar yet functionally distinct. The model appears to learn more about geometric matching between humans and objects than a deeper, semantic understanding of functional accessibility.
* This method relies on a single standard human pose (H0) as a fixed baseline for flow prediction. However, significant variations in human morphology and dimensions directly influence how individuals interact with objects. Can the method be generalized to different people's post H0?  How would the predictability and reliability change if the standard pose H0 represented individuals of varying body types?

**Questions:**

See the Weaknesses

---

> ### Author Response · Authors · 2025-11-19
> **(1/3) Thank you, please see below for response. We have added new scalability results in the revised manuscript (**see highlighted text and new Appendix Q: Scalability**)**
>
> We sincerely thank Reviewer Y1sC for the constructive and detailed review. We appreciate the recognition of the **dense diffused flow** representation, the **point-cloud–native** training/inference (no watertight meshes), and the empirical **robustness** to noisy, partial real scans. This also echoes other reviewers’ comments on the **novel contribution** of flow formulation in HOI affordance understanding. We now address the reviewer's questions below.
>
> > On the choice of upstream data generator and current data source.
>
> **Current training data selection scheme.** For each object in the training dataset, we generate 100 HOI sequences. Each sequence contributes 200 frames, taken as the middle 200 key frames of the generated sequence. We treat each frame as an independent training datapoint (after mesh $\rightarrow$ FPS $\rightarrow$ paired point-cloud preprocessing), which yields a large, diverse, and balanced set without additional annotation. We augment the data by random occlusion and rotation (see **Appendix E**).
>
> **Why CHOIS.** We chose CHOIS for diversity (broad object coverage, varied motion styles) and efficiency (every frame in a sequence is usable as a datapoint), which makes it a strong upstream source for our point-based learning of dense diffused flows. While there are more recent works on HOI data synthesis, CHOIS was the SOTA by the time we wrote this paper. However, as we shall see in the next section, H2OFlow scales with more diverse data generated by more recent HOI synthesis work.

---

> ### Author Response · Authors · 2025-11-19
> **(2/3) **Please see Appendix Q of the revised manuscript for new results and figures for this section****
>
> > On semantic control and decoupling
>
> We thank the reviewer for raising this question. The current version of H2OFlow was trained on CHOIS-based synthesis, which tends to over-represent lifting/moving activities. However, our pipeline is **model-agnostic** to the HOI source. To overcome this issue, we propose **two complementary solutions**, which are provided in **Appendix Q: Scalability**. As new experiments suggest, H2OFlow does scale well with the diversity of the synthesized HOI meshes, provided by more recent 3D generative models. With this scalability, H2OFlow handles **usage-centric affordances** well. We summarized some key points below:
>
> - **Scale the training distribution to include usage HOIs (e.g., sitting/leaning/resting).** We augment the synthetic HOI pool with recent 3D HOI generation models that explicitly produce usage-centric human–object interactions such as InteractAnything [1], HOI-PAGE [2], and PICO [3]. All generators are standardized by our existing preprocessing: mesh $\rightarrow$ FPS subsampling $\rightarrow$ paired point clouds, identical to our current pipeline. No watertight meshes or normals are required downstream. By adding more training data, we effectively expand the actions used: “sit on/lean on/place on/backrest/armrest.” H2OFlow’s pipeline (mesh $\rightarrow$ FPS $\rightarrow$ point clouds) and training **do not change**; only the HOI source distribution broadens.
>   - See **Appendix Q.1 (Scaling via Data Augmentation)** for the recipe and chair case study visualization.
> - It is worth noting here that H2OFlow actually **already learns some semantic information of the objects** based on each part’s intended usage, as a **byproduct** of the per-point embedding learning in DiT. As shown in the new **Figure 10 in Appendix Q**, where we perform t-SNE visualization of the per-point embeddings using only CHOIS data, we observe that vertices from *semantically equivalent parts* form tight, part-consistent clusters, where each cluster roughly indicates the usage of the chair when it is being moved.
>   - Specifically, for a chair, clusters concentrate on the outer edges of the seat back and the seat surface, reflecting “lifting/moving'' usage dominant in CHOIS data and agreeing with what we have seen in the affordance visualization before augmentation in the new **Figure 9**. After augmenting the training dataset with more sources, H2OFlow actually begins to learn more geometric-semantic information of the object (see **Appendix Q.2: Learned Geometric-Semantic Information** for details): for the chair case study, we observed that a new usage of the chair seat emerged from the augmented dataset, indicating that the **geometric-semantic information learned also scales** with the training data diversity, and more importantly, that that H2OFlow captures a local geometric-semantic space rather than memorizing specific meshes.
> - **Add language conditioning to steer flows at inference.** Beyond dataset diversification, we can *condition* the dense-flow generator on a textual intent (“sit on the chair''). Please kindly note that this is more of a natural next step for H2OFlow, but we *preview some preliminary results* here to demonstrate the scalability of H2OFlow. The text conditioning steers sampling toward usage-consistent interactions at test time, while keeping H2OFlow’s point-based formulation intact. We add a lightweight text cross-attention into each DiT block (Option A), enabling prompts like “sit on chair” to bias sampled flows toward seat occupancy and torso/backrest alignment without changing affordance inference. This implementation requires **minimal architectural change**, as seen in **Appendix Q.3 (Prompt-Conditioned Dense Diffused Flows)**, where we document the change of implementation in PyTorch code (**Fig. 11 and Fig. 12**). In the new **Fig. 13**, we use the same chair case study example as above but test it on the text-conditioned H2OFlow.
>   - As one can observe, with the prompt conditioning, the two different usages of the chair yield really different affordances. The contact now is not bimodal between hands and hips, and is now concentrated on one body part based on the usage. Similarly, for the orientation affordance, we observe high concentration on the hips when sitting and on the arms when moving. More interestingly, we see a sitting silhouette for sitting and a standing silhouette for moving.
>
> [1] Cai, Eric, et al. "Non-rigid Relative Placement through 3D Dense Diffusion." (CoRL, 2024).
>
> [2] Zhang, Jinlu, et al. "InteractAnything: Zero-shot Human Object Interaction Synthesis via LLM Feedback and Object Affordance Parsing." (CVPR, 2025)
>
> [3] Li, Lei, and Angela Dai. "HOI-PAGE: Zero-Shot Human-Object Interaction Generation with Part Affordance Guidance." (2025).
>
> [4] Cseke, Alpár, et al. "PICO: Reconstructing 3D people in contact with objects." (CVPR, 2025).

---

> ### Author Response · Authors · 2025-11-19
> **(3/3)**
>
> > On the choice of a single human pose ($H_0$)
>
> **Design rationale.** Using a fixed canonical human pose ($H_0$) is an anchor that turns affordance prediction into an **object-conditioned** task: the model predicts dense flows from the same anchor toward interaction configurations conditioned on the object. Because the representation is point-cloud–relative, translational equivariance is guaranteed by construction; the model need not first infer the user’s current pose. This is a design choice that simplifies learning and is **not a data limitation**.
>
> **Ablation with multiple anchors.** We trained four models with distinct anchors (*stand, sit, crouch, T-pose*) on a smaller-scale set; at test time we take best-of-4. The best-of-4 ensembling showed no statistically significant improvement over a single anchor while substantially increasing training cost. This suggests the flow predictor is effectively agnostic to the starting pose; performance is not bottlenecked by the choice of canonical anchor.
>
> **Varying morphology.** We agree that morphology can influence clearances in **tight geometries** (e.g., narrow handles, slots) or tasks requiring fine ergonomic fit. These cases are **comparatively rare** in current HOI synthesis literature, which skew toward *neutral body shapes*, but they motivate a targeted extension. Without changing affordance inference, we can add morphology with small, modular changes.
>   - For example, we could implement shape augmentation at data time, where we sample SMPL shape parameters $\beta$ during HOI synthesis. Then, for each generated HOI clip (poses + object), re-pose the human with the sampled $\beta$ using the same joint angles. Run a quick collision check and resample if the clip becomes invalid (e.g., interpenetration with the object). In this way, we can keep the same mesh, FPS, point-cloud pipeline.
>
> We once again thank the reviewer for pointing out the corner cases of varying morphology, and we will implement this in the next version of H2OFlow.
>
> These changes strengthen the paper while preserving its core contribution: **learning comprehensive, robust 3D affordances from point clouds via dense diffused flows**, scalable to new generators, tasks, and semantic controls. We hope these revisions clarify our contributions and demonstrate the robustness, generalizability, and scalability of H2OFlow. Please kindly consider raising the score if your concerns and questions are addressed.

---

> ### Comment · Reviewer_Y1sC · 2025-11-24
> **Response to the rebuttal**
>
> Thanks for the authors' reply, which has resolved my questions, and I still maintain a positive attitude toward accepting this paper.

---

> > ### Author Response · Authors · 2025-11-24
> >
> > Thank you so much for acknowledging our rebuttal and advocating for our acceptance!

---

### Official Review · Reviewer_LAsr · 2025-10-26

**Soundness:** 2
**Presentation:** 1
**Contribution:** 3
**Rating:** 4
**Confidence:** 4

**Summary:**

This paper presents H2OFlow, a framework for learning comprehensive 3D human-object interaction (HOI) affordances that goes beyond simple contact-based analysis. The authors address a key limitation in current affordance understanding methods, which typically focus only on contact regions while neglecting other critical aspects of human-object interactions, such as spatial orientation preferences and occupancy patterns.

The core motivation stems from the observation that human-object interactions involve rich 3D spatial relationships—for example, humans maintain characteristic distances and orientations when interacting with objects. Building on the concept of "comprehensive affordance" from Kim et al. (2024), which models probabilistic distributions over 3D spatial and orientational relations, the authors propose a novel approach that operates directly on point clouds rather than requiring mesh-based representations or 2D-to-3D uplifting.

Key contributions include:

A point-cloud-based affordance representation that captures both explicit contact and implicit non-contact interaction patterns from raw point cloud inputs
A synthetic data generation pipeline leveraging 3D generative models and dense diffusion flows (inspired by Eisner et al. 2022) to learn flexible affordances without manual annotations
A probabilistic formulation that operates directly on human-object point cloud pairs, eliminating the dependency on watertight meshes
The authors claim that H2OFlow generalizes effectively to real-world objects and outperforms existing methods that rely on manual annotations or mesh-based representations. The framework uses only synthetic data generated from 3D generative models, potentially addressing the scalability issues associated with labor-intensive dataset creation in this domain.

**Strengths:**

Originality:
1. This paper introduces a novel paradigm shift from manual contact annotation to direct learning from point cloud datasets.
2. The integration of dense diffusion flows with 3D generative models represents an innovative approach to synthetic data generation that eliminates dependency on high-quality mesh inputs.
3. It extends beyond the traditional binary contact-based affordance definition by incorporating spatial orientation and occupancy patterns, providing a more comprehensive and nuanced understanding of human-object interactions.

Quality: This paper achieves better qualitative and quantitative results on affordance learning compared with previous methods.

Clarity: The training method, inference time method, and dataset acquisition methods are clear.

Significance: This work addresses a critical challenge in computer vision, robotics, and AI by providing a more scalable and comprehensive approach to affordance learning. The elimination of manual annotation requirements and mesh dependencies has substantial practical implications for real-world applications. The comprehensive affordance representation (contact + orientation + spatial occupancy) offers significantly more flexibility for downstream tasks and could enable more sophisticated robotic manipulation and scene understanding capabilities. The potential impact extends beyond affordance learning to broader areas of 3D scene understanding and human behavior modeling.

**Weaknesses:**

1. Although it uses the comprehensive affordance representations for better affordance definition, all three representations are proposed in previous methods, lacking original definitions and contributions. Additionally, there is no ablation study on whether the two additional representations actually yield better results for affordance learning.
2. Compared with only one previous baseline COMA, lacking experiments.
3. Most of the main figures are in the supplementary, indicating that the paper is not well organized.

**Questions:**

1. In Section 4.1, in the first sentence, the pretrained 3D model is used to generate meshes. However, in the last sentence of the same paragraph, the outputs are videos. So what does the pretrained model actually output?
2. Why use FPS to sample points? Have you tried other sampling policies?
3. For unseen objects, can you generate reasonable affordances for this object? For example, in Figure 5, most of the training data is human hands interacting with the object, and then for the unseen object, the chair, it also gives information about how hands interact with it. However, for chairs, maybe "Hips sitting on the chair" is a more common affordance. How can the method handle this situation?
4. Are the additional affordance representations (except contact) useful? Where is the ablation study about this?

---

> ### Author Response · Authors · 2025-11-19
> **(1/3) Thank you, please see below for response. We have changed the figure locations and added new ablations and scalability results in the revised manuscript (**see highlighted text, new experiments in Section 5,and new Appendix Q: Scalability**)**
>
> We sincerely thank Reviewer LAsr for the constructive feedback and appreciation of our contributions. We agree and emphasize that our paradigm **removes manual contact labels and operates directly on point clouds**; dense diffused flows let us learn rich HOI structure from synthetic 3D generators without watertight meshes. This also echoes Reviewer Y1sC’s comment on the **elegance of the flow formulation**. We respond to the questions raised below.
>
> > On previous work's affordance formulation
>
> While previous work proposed a similar set of comprehensive affordances, H2OFlow introduces four fundamental technical advances over previous work:
> - **More generalizability and flexibility**. Most fundamentally, previous work directly uses reconstructed 3D inputs and has no learned components in their pipeline. While previous work lays out comprehensive affordances in a nice way, the lack of learning-based methods limits its generalizability and flexibility when it comes to unseen objects (as we noted in the quantitative results of the paper).
> - **Point-cloud-based affordance learning paradigm with dense diffused flows as an effective intermediate representation.** In previous work, affordances are inferred from reconstructed meshes instead of learned flow representations, which lacks the generalizability to real-world visual inputs. In H2OFlow, with flows, no watertight meshes or surface normals are needed, which tend to be noisy in real world. Previous work struggles to generalize to unseen objects and reconstructed meshes from noisy point clouds, while H2OFlow performs well in both cases.
> - **Diffusion-based multi-modal dense-flow predictor based on per-point encoding.** This learning paradigm handles intrinsic ambiguity due to multi-modality and also learns to reason about geometric information on different regions (local information) of the object-human interaction. With dense diffused flows, H2OFlow's pipeline provides a new method for modeling human pose with a more flexible representation. At the same time, this representation **sidesteps the need for normal vectors** from meshes (Eq. 5 and 6), which are costly to compute for real-world applications, while achieving better results.
> - **Cross-attention aggregation & partial-scan robustness.** We improve the affordance aggregation via learned cross-attention weights (see ablations in Table 1). During training, we apply random rotation to the objects and random occlusion as augmentation in order to ensure robustness to real-world variability (Appendix E), which ensures the robustness to occlusion in the real world, as opposed to previous work that requires a clean mesh of the object. In Table 6 of the paper, we evaluate H2OFlow's sensitivity to occlusion on OMOMO test objects and show that the performance loss due to occlusion and partial observability is minimal, indicating robustness to commodity depth cameras or monocular depth-completion pipelines.
>
> > On the chosen previous baselines
>
> We focused on COMA because it is, to our knowledge, the **only** prior work that attempts **comprehensive** affordances beyond contact. That said, we also **compare to contact-only** baselines (including COMA’s contact maps and additional contact-centric methods) in the appendix as recommended by prior work; see the contact-only comparisons referenced from the main text (**Appendix J** mentioned in our results narrative). We have made these more explicit in the revision to address discoverability.
>
> > Figures mostly in supplementary, organization.
>
> We appreciate the reviewer’s helpful comments on presentation clarity. Some figures were deferred to the appendix due to space constraints, but they are now in the main paper thanks to the extra page allowed. In the revised version, where changes were highlighted, we have reorganized the figures in Section 3 and Section 4 to improve readability and flow. Specifically:
>
> - We have moved the overview figure (previously in the appendix) into the main paper as Figure 2, now directly referenced at the beginning of Section 3 for better conceptual grounding.
> - Section 4 now references the new Figure 2 when we describe our methods in detail.
> We believe these revisions substantially improve the paper’s organization and help readers better follow our formulation and method.
>
> > On the 3D generative model output
>
> The pretrained 3D HOI generator outputs **mesh sequences** (temporally synchronized human-object meshes). These sequences can be rendered as videos for visualization, which caused the wording confusion. We have clarified the text: “the model outputs **mesh sequences**.”
>
> **(REBUTTAL CONTINUED IN THE NEXT SECTION...)**

---

> ### Author Response · Authors · 2025-11-19
> **(2/3) **Please see Appendix Q of the revised manuscript for new results and figures for this section****
>
> > On FPS Sampling
>
> We use Furthest Point Sampling (FPS) because it yields stable, uniformly covering subsets across partial scans and supports our per-point dense-flow/attention computations efficiently. FPS also avoids bias toward dense local geometry and works well with the cross-attention formulation in DiT [1]. We also tried uniform sampling during training, but we observed a **less stable** training process.
>
> > Usage-centric affordances
>
> We thank the reviewer for raising this question. The submitted version of H2OFlow was trained on CHOIS-based synthesis, which tends to over-represent lifting/moving activities. To overcome this issue, we propose **two complementary solutions**, which are provided in **Appendix Q: Scalability**. As new experiments suggest, H2OFlow does **scale well with the diversity** of the synthesized HOI meshes, provided by more recent 3D generative models. With this scalability, H2OFlow handles usage-centric affordances well. We summarized some key points below:
> - **Scale the training distribution to include usage HOIs (e.g., sitting/leaning/resting).** We augment the synthetic HOI pool with recent 3D HOI generation models that explicitly produce usage-centric human–object interactions such as InteractAnything [2], HOI-PAGE [3], and PICO [4]. All generators are standardized by our existing preprocessing: mesh $\rightarrow$ FPS subsampling $\rightarrow$ paired point clouds, identical to our current pipeline. No watertight meshes or normals are required downstream. By adding more training data, we effectively expand the actions used: “sit on/lean on/place on/backrest/armrest.” H2OFlow’s pipeline (mesh $\rightarrow$ FPS $\rightarrow$ point clouds) and **training does not change**; only the HOI source distribution broadens.
>   - See **Appendix Q.1 (Scaling via Data Augmentation)** for the recipe and chair case study visualization.
>   - After augmenting the training dataset with more sources, H2OFlow actually begins to learn more geometric-semantic information of the object (see **Appendix Q.2: Learned Geometric-Semantic Information** for details): for the chair case study, we observed that a new usage of the chair seat emerged from the augmented dataset from the t-SNE visualization of the per-point embeddings, indicating that the **geometric-semantic information learned also scales** with the training data diversity.
> - **Add language conditioning to steer flows at inference.** Beyond dataset diversification, we can *condition* the dense-flow generator on a textual intent (“sit on the chair''). Please kindly note that this is more of a natural next step for H2OFlow, but we *preview some preliminary results* here to demonstrate the scalability of H2OFlow. The text conditioning steers sampling toward usage-consistent interactions at test time, while keeping H2OFlow’s point-based formulation intact. We add a **lightweight text cross-attention** into each DiT block, enabling prompts like “sit on chair” to bias sampled flows toward seat occupancy and torso/backrest alignment without changing affordance inference.
>   - This implementation requires **minimal architectural change,** as seen in **Appendix Q.3 (Prompt-Conditioned Dense Diffused Flows)**, where we document the change of **implementation in PyTorch code (Fig. 11 and Fig. 12)**.
>   - In the new **Fig. 13**, we use the same chair case study example as above but test it on the text-conditioned H2OFlow. As one can observe, with the prompt conditioning, the two different usages of the chair **yield really different affordances**. The contact now is not bimodal between hands and hips, and is now concentrated on one body part based on the usage. Similarly, for the orientation affordance, we observe high concentration on the hips when sitting and on the arms when moving. More interestingly, we see a sitting silhouette for sitting and a standing silhouette for moving.
>
> These two additions directly address the usage-specific scenarios of HOI affordances by (i) changing what the model learns from, and (ii) allowing the user/task to steer what it generates—both while preserving our point-cloud flow formulation.
>
> [1] Cai, Eric, et al. "Non-rigid Relative Placement through 3D Dense Diffusion." (CoRL, 2024).
>
> [2] Zhang, Jinlu, et al. "InteractAnything: Zero-shot Human Object Interaction Synthesis via LLM Feedback and Object Affordance Parsing." (CVPR, 2025)
>
> [3] Li, Lei, and Angela Dai. "HOI-PAGE: Zero-Shot Human-Object Interaction Generation with Part Affordance Guidance." (2025).
>
> [4] Cseke, Alpár, et al. "PICO: Reconstructing 3D people in contact with objects." (CVPR, 2025).
>
> **(REBUTTAL CONTINUED IN THE NEXT SECTION...)**

---

> ### Author Response · Authors · 2025-11-19
> **(3/3) **Please see Section 5 and Appendix P of the revised manuscript for new results and details for this section****
>
> > On the usefulness of new affordances
>
> We now include a new ablation isolating each term’s contribution (**Contact-only C, Contact+Orientational C+O, Contact+Spatial C+S, Full C+O+S, plus shuffle controls, where we keep C but shuffle O and S**) while holding the flow sampler fixed. The detailed description of this new experiment is provided in **Section 5: Affordance Types Ablation** of the main text and **Appendix P: Are Non-Contact Affordances Useful?** We summarize some key points below.
>
> - We design two downstream HOI inference tasks on unseen OMOMO objects:
>   - HOI Region Retrieval: Given an object query point $o_j$, rank human points by $\phi_{ij}$; compute mAP@{1,5,10} against GT contact points, where $\phi\_{ij} = \lambda_c \widehat{c}\_{ij} + \lambda_o \widehat{R}\_{ij} + \lambda_s \widehat{S}\_{ij},$
>   - Pose Selection: Given sampled HOI hypotheses per object, select $\arg\max_{k}\sum_{i,j}\phi^{(k)}_{ij}$. Report Top-$k$ accuracy vs. GT pose clusters, collision rate with object, and contact leakage that measures contacts on implausible parts.
> - Quantitative results of the above metrics are shown in the newly added **Table 2** and statistical testing was conducted in **Appendix P**.
>   - Results suggest that against $\mathbf{C}$, $\mathbf{C{+}O}$ and $\mathbf{C{+}S}$ significantly improve the metrics,  and $\mathbf{C{+}O{+}S}$ yields further gains. Shuffled controls eliminate these improvements, confirming that structured orientational and spatial affordances indeed improve performance for affordance learning, not merely because of additional feature capacity.
>
> These additions strengthen the paper’s empirical case while preserving its core contributions: **learning comprehensive 3D affordances from point clouds via dense diffused flows**, with robust generalization to real-world inputs. We hope these revisions clarify our contributions and demonstrate the robustness, generalizability, and scalability of H2OFlow. Please kindly consider raising the score if your concerns and questions are addressed.

---

> ### Author Response · Authors · 2025-11-25
> **Kindly seeking feedback from reviewer**
>
> We sincerely thank you for your thoughtful review. As the discussion period draws to a close, we would appreciate your feedback to confirm whether our replies have addressed your concerns. If you have any remaining questions, we are happy to provide further clarification. If our responses have resolved your concerns, we would be deeply grateful if you could consider raising the score. Thank you again for your time and effort during the review process.

---

> ### Author Response · Authors · 2025-11-27
> **Kindly seeking feedback and acknowledgement**
>
> Dear Reviewer LAsr, thank you for your review. If you have any remaining questions, we are happy to provide further clarification. If our responses have resolved your concerns, we would be deeply grateful if you could consider raising the score. Thank you again for your time and effort during the review process.

---

### Official Review · Reviewer_qngF · 2025-10-27

**Soundness:** 2
**Presentation:** 2
**Contribution:** 2
**Rating:** 4
**Confidence:** 3

**Summary:**

The paper studies the problem of understanding the affordance of human object interaction. To this end, the authors propose a representation that captures not only contact affordance, but also orientational affordance and spatial affordance. The authors propose a pipeline to generate synthetic data using 3D generative models and employ a dense 3D-flow-based representation. The authors claim that, through extensive quantitative and qualitative experiments, the effectiveness and practical utility of the learned affordances on both synthetic datasets and real-world data are demonstrated.

**Strengths:**

1. The introduction of the affordance representation that captures both explicit contact and implicit non-contact interaction patterns is novel to me.

2. The problem the paper studies is important. If we want to move to spatial and physical AI in the future, it is important to understand human-object affordance.

**Weaknesses:**

1. The organisation of the paper needs to be improved. The overview figure as referred in Section 3 is very important for the understanding of the problem formulation, but the authors put it in the appendix. In Section 4, the authors describe a lot about their method, and it would be a lot better if there is a figure to demonstrate the whole method.

2. Insufficient real-world results. The authors claim that their method can be well generalised to unseen real-world objects. It would be more convincing if the authors can provide quantitative results, given that at the moment there is only qualitative results.

**Questions:**

Please see the weaknesses part of my review. I would suggest the authors to address my concerns in the rebuttal.

---

> ### Author Response · Authors · 2025-11-19
> **Thank you, please see below for response. We have changed the figure locations and added quantitative real-world results in the revised manuscript (**see highlighted text and new Appendix O**)**
>
> We sincerely thank Reviewer qngF for the constructive feedback and appreciation of our contributions. We are glad that the reviewer found our proposed affordance representation **novel and recognized the importance of studying human-object affordances** for advancing spatial and physical AI. This also resonates with Reviewer LAsr’s point on the significance of H2OFlow. We fully agree that understanding not only contact but also orientational and spatial affordances is crucial for enabling AI systems to reason about physical interactions in 3D environments. We respond to the reviewer's questions below.
>
> > On Paper Organization and Figures
>
> We appreciate the reviewer’s helpful comments on presentation clarity. Some figures were deferred to the appendix due to space constraints, but they are now in the main paper thanks to the extra page allowed. In the revised version, where changes were highlighted, we have reorganized the figures in Section 3 and Section 4 to improve readability and flow. Specifically:
> - We have moved the overview figure (previously in the appendix) into the main paper as Figure 2, now directly referenced at the beginning of Section 3 for better conceptual grounding.
> - Section 4 now references the new Figure 2 when we describe our methods in detail.
> We believe these revisions substantially improve the paper’s organization and help readers better follow our formulation and method.
>
> > On Quantitative Real-World Evaluation
>
> We thank the reviewer for pointing out the need for quantitative validation on real-world data. In the revision, we have added a new quantitative experiment on six real-world object scans, aligned with simulated counterparts from OMOMO and BEHAVE datasets.
>
> To address the absence of ground truth affordance labels, we designed a label-free quantitative protocol comparing SMPL human poses generated by different methods against CHOIS synthetic reference poses. The detailed metrics and results are in **Appendix O: Real-World Quantitative Results**, but the key points are summarized here.
> - We select six real-world objects that have similar simulated counterparts in the OMOMO and BEHAVE datasets. Each real object point cloud is aligned to a canonical object frame using ICP with its closest simulated counterpart.
> - We compute set-to-set distances in joint space (Minimum Matching Distance, Coverage@15cm, and Frechet Pose Distance) between predicted and reference SMPL distributions.
>   - First, Minimum Matching Distance (MMD) measures the average minimum per-sample joint distances $d(S,S')=\frac{1}{J}\sum_{j=1}^{J}||J_j(S)-J_j(S')||_2$.
>     - Then $\text{MMD}(\mathcal{S},\mathcal{S}\_{\text{ref}}) =\frac{1}{|\mathcal{S}|} \text{min}\_{S'\in\mathcal{S}_{\text{ref}}}d(S,S').$
>   - Next, Coverage measures the percentage of candidate set that covers the modes of reference set: $\text{COV}_\varepsilon =
> \frac{|\{S'\in\mathcal{S}\_{\text{ref}}:\exists S\in\mathcal{S},d(S,S')\le\varepsilon\}|}
> {|\mathcal{S}\_{\text{ref}}|},$
>   - Lastly, Frechet Pose Distance (FPD) measures the distributional similarity between the prediction sets:
> $\text{FPD} = \|\mu_c-\mu_r\|\_2^2 + \mathrm{Tr}(\Sigma\_c+\Sigma\_r - 2(\Sigma\_c^{1/2}\Sigma\_r\Sigma\_c^{1/2})^{1/2}), $
>
> where $J_j(S)$ denotes the 3D joint positions of pose $S$, and $(\mu_c,\Sigma_c)$, $(\mu_r,\Sigma_r)$
> are the means and covariances of joint vectors from the candidate and reference sets, respectively.
> Lower MMD and FPD and higher COV$_\varepsilon$ indicate closer alignment between predicted and synthetic interaction distributions.
>
> Results show that H2OFlow achieves significantly lower MMD/FPD and higher coverage than COMA, confirming its strong generalization to real-world objects without manual annotation. These new results are reported in **Appendix Section O (Real-World Quantitative Results)** and summarized in **Table 6 of the revised manuscript**, strengthening the empirical validation of our claims.
>
>
>
>
> We hope these revisions clarify our contributions and demonstrate the robustness and generalizability of H2OFlow. Please kindly consider raising the score if your concerns and questions are addressed.

---

> ### Author Response · Authors · 2025-11-25
> **Kindly seeking feedback from reviewer**
>
> We sincerely thank you for your thoughtful review. As the discussion period draws to a close, we would appreciate your feedback to confirm whether our replies have addressed your concerns. If you have any remaining questions, we are happy to provide further clarification. If our responses have resolved your concerns, we would be deeply grateful if you could consider raising the score. Thank you again for your time and effort during the review process.

---

> ### Author Response · Authors · 2025-11-27
> **Kindly seeking feedback and acknolwdgement**
>
> Dear Reviewer qngF, thank you for your review. If you have any remaining questions, we are happy to provide further clarification. If our responses have resolved your concerns, we would be deeply grateful if you could consider raising the score. Thank you again for your time and effort during the review process.

---

### Author Response · Authors · 2025-11-20
**Paper Revision and Rebuttal Summary: High-level results in textbox, details and figures in revised manuscript pdf.**

We sincerely appreciate the reviewers' insightful comments and questions. We have responded to each raised question below in the text boxes and changed the manuscript pdf with added experiments and more figures. The changes are highlighted in the updated pdf and below is a list of changes made to the main paper and appendix:
- Moved the system figure (Figure 2) from the appendix to the main text.
  - This aims to improve the presentation of the paper.
- Clarified upstream data format in Section 4.1.
- Added reference to the system figure in Section 4.2.
- Emphasized the comparison with contact-only baselines in Section 5.1.
- Updated Figure 4 with colormaps.
- Added Affordance Types Ablation in Section 5.2
  - This confirms that structured orientational and spatial affordances indeed improve performance for affordance learning, not merely because of additional feature capacity.
- Added real-world quantitative results in Appendix O.
  - Appendix O.1 defines metrics for real-world experiments and shows quantitative results.
  - Appendix O.2 ablates on the denoising aggressiveness in the real world and shows H2OFlow's robustness to noise level.
- Added experiment details and statistical testing results for Affordance Types Ablation in Appendix P.
  - Defined two downstream HOI tasks for comparison.
  - Quantitatively demonstrated the benefits of using affordances beyond just contacts.
- Added scalability experiments in Appendix Q.
  - Appendix Q.1 shows H2OFlow scales with data sources: we can eliminate the skewness in single data sources by training on multiple data sources, without changing the pipeline or flow representations. We show qualitative results illustrating that H2OFlow learns more usage-centric affordances from diverse data sources without architectural change.
  - Appendix Q.2 shows that H2OFlow learns geometric-semantic information of the object, and this information also scales with more data: t-SNE visualization of object per-point embedding shows that learned embeddings cluster roughly by usage, and new usage emerges after augmenting with more data.
  - Appendix Q.3 shows that, with minimal architectural change, we can condition H2OFlow outputs on textual input to reflect multiple use cases. We show that a simple change in implementation is able to support this extension.

---

### Author Response · Authors · 2025-11-29
**Thank you. Summary for ACs**

We wish to extend our appreciation to the ICLR committee for their diligent work in upholding the standards of the review process. To aid the newly assigned Area Chair, and, in light of their potentially increased workload, we have compiled this summary of the key points from our prior interactions with the reviewers.

All reviewers have highlighted the technical contributions of H2OFlow, especially with respect to:

- The **elegance** of the dense flow representation to bypass the need for meshes.
- The **robust performance** across tasks and metrics.
- The problem that H2OFlow aims to address is a **critical problem**.
- The **potential impact** of the work is significant.

In the discussion period, we revised the draft (all changes are highlighted) to add more experiments and addressed the reviewers' questions by:

- Appendix O: Providing real-world quantitative results.
- Appendix P: Providing ablations demonstrating the efficacy of using more than just contact in downstream affordance learning tasks.
- Demonstrating that the modular design of H2OFlow is easily extendable, especially via:
  - Appendix Q.1: Augmenting the training dataset with diverse sources for usage-centric affordances.
  - Appendix Q.3: Augmenting the flow predictor with language conditioning to allow for prompt-conditioned affordances.
  - Appendix Q.2: Illustrating that the semantic-geometric information learned scales with data as well, using t-SNE visualization.

As indicated by the reviewers who replied to our rebuttal during the discussion period, our new experiments in the rebuttal **have addressed their questions**, and they maintain a **positive attitude** toward accepting this work. We also added new symmetry-enforced evaluations to respond to Reviewer TEPZ's point that symmetry in affordance is often overlooked and is worth studying.

We once again extend our deepest appreciation to the organizers and ACs for their time and effort in ensuring the fairness of ICLR reviews.

---

### Meta-Review · Area_Chair_rLtG · 2026-01-11

**Summary:**

This paper tackles 3D human–object interaction (HOI) affordance understanding and argues that affordances should go beyond contact regions to also capture preferred orientation and spatial occupancy around objects. To avoid expensive HOI annotation, the authors propose H2OFlow, a framework that trains purely on synthetic HOI data generated from a pretrained 3D generative model. The paper is novel and timely, and reviewers broadly agree the direction—learning comprehensive 3D affordances from synthetic HOI without manual labels—is impactful. There are concerns on insufficient real-world quantitative validation and incomplete analysis. After carefully reading the paper, review and author responses, the AC agrees with the reviewers on accepting the paper.

**Reviewer Concerns:**

see Summary

**Reviewer Scores:**

see Summary

---

### Decision · Program_Chairs · 2026-01-26

Accept (Poster)